# Surface photogalvanic effect in Ag$_2$Te

Xiaoyi Xie[1,2,10], Pengliang Leng[1,2,10], Zhenyu Ding [3,10], Jinshan Yang [4,10], Jingyi Yan[4], Junchen Zhou[1,2], Zihan Li [1,2], Linfeng Ai[1,2], Xiangyu Cao[1,2], Zehao Jia[1,2], Yuda Zhang[1,2], Minhao Zhao[1,2], Wenguang Zhu [3,5,6], Yang Gao [5] ✉, Shaoming Dong[4] & Faxian Xiu [1,2,7,8,9] ✉

The bulk photovoltaic effect (BPVE) in non-centrosymmetric materials has attracted significant attention in recent years due to its potential to surpass the Shockley-Queisser limit. Although these materials are strictly constrained by symmetry, progress has been made in artificially reducing symmetry to stimulate BPVE in wider systems. However, the complexity of these techniques has hindered their practical implementation. In this study, we demonstrate a large intrinsic photocurrent response in centrosymmetric topological insulator Ag$_2$Te, attributed to the surface photogalvanic effect (SPGE), which is induced by symmetry reduction of the surface. Through diverse spatially-resolved measurements on specially designed devices, we directly observe that SPGE in Ag$_2$Te arises from the difference between two opposite photocurrent flows generated from the top and bottom surfaces. Acting as an efficient SPGE material, Ag$_2$Te demonstrates robust performance across a wide spectral range from visible to mid-infrared, making it promising for applications in solar cells and mid-infrared detectors. More importantly, SPGE generated on low-symmetric surfaces can potentially be found in various systems, thereby inspiring a broader range of choices for photovoltaic materials.

The bulk photovoltaic effect (BPVE) is a direct current (dc) that occurs in non-centrosymmetric materials under illumination[1], which was discovered in ferroelectrics in the 1970s[2,3]. Due to its potential to surpass the Shockley-Queisser limit and achieve photovoltage exceeding the bandgap, BPVE regained attention as an alternative to $p$–$n$ junction for charge separation in solar power converting, photo-sensing, and other optoelectronic applications[4–6]. Meanwhile, BPVE has also attracted interest for its microscopic mechanisms associated with topological physics and quantum geometry, which mainly stem from the shift current theory[7–9]. Recently research on BPVE has expanded to a wide range of materials, including traditional ferroelectric materials[5,10],

perovskites[11], organic compounds[12], transition metal dichalcogenides[13], topological insulators[14], Weyl semimetals[15], and so on.

Symmetry plays a crucial role in the photocurrent generation of BPVE. The second-order BPVE under linear illumination can be described as[1]:

$$j_q = \beta_{qrs} e_r e_s^* I \qquad (1)$$

Where $q$, $r$, and $s$ are Cartesian coordinates, $j_q$ is the photocurrent density, $e_r$ and $e_s^*$ are the light polarization unit vectors, $I$ is the light intensity, and $\beta_{qrs}$ is a third-rank tensor named the BPV coefficient. The

[1]State Key Laboratory of Surface Physics and Department of Physics, Fudan University, Shanghai 200433, China. [2]Shanghai Qi Zhi Institute, 41st Floor, AI Tower, No. 701 Yunjin Road, Xuhui District, Shanghai 200232, China. [3]International Center for Quantum Design of Functional Materials (ICQD), Hefei National Research Center for Physical Sciences at the Microscale, University of Science and Technology of China, Hefei 230026, China. [4]State Key Laboratory of High Performance Ceramics and Superfine Microstructure, Shanghai Institute of Ceramics, Chinese Academy of Science, Shanghai 200050, China. [5]Department of Physics, University of Science and Technology of China, Hefei 230026, China. [6]Hefei National Laboratory, University of Science and Technology of China, Hefei 230088, China. [7]Institute for Nanoelectronic Devices and Quantum Computing, Fudan University, Shanghai 200433, China. [8]Zhangjiang Fudan International Innovation Center, Fudan University, Shanghai 201210, China. [9]Shanghai Research Center for Quantum Sciences, Shanghai 201315, China. [10]These authors contributed equally: Xiaoyi Xie, Pengliang Leng, Zhenyu Ding, Jinshan Yang. ✉e-mail: ygao87@ustc.edu.cn; Faxian@fudan.edu.cn

symmetry constraint for BPVE can be extracted straightforwardly by applying symmetry operations to Eq. (1). For instance, once a system has an inversion center, the right side of the equation will maintain its sign after an inversion operation while the left side will change its sign. Consequently, $\beta$ and then $j$ have to be zero in this system. As a result, inversion symmetry breaking is the well-known fundamental constraint of BPVE. Nevertheless, some other symmetry operations also lead to the vanishing response, such as out-of-plane twofold rotation symmetry with a geometry of in-plane photocurrent and normal incidence[16]. It's necessary to perform concrete group theory analysis to determine the presence of $\beta$. Furthermore, recent experimental studies have demonstrated that certain structures satisfying the symmetry requirements according to Eq. (1) exhibit poor BPVE response. However, a significant improvement is observed when the symmetry is further reduced in these systems[13,17].

Since the essentiality of symmetry, novel approaches involving artificially reduced symmetry have been proposed, such as the strain gradient-induced flexo-photovoltaic effect achieved through an AFM tip[18] or 2D materials stacking technique[19], strain-induced piezo-photovoltaic effect[13], and asymmetric stacked structures[20,21]. These leaps make it possible to introduce BPVE in various materials with excellent optoelectronic properties, like transition metal dichalcogenide, black phosphorus, and graphene, regardless of their prime symmetry. However, the technical implementation of these methods generally involves complex procedures, which hinder their practical applications.

Surface photogalvanic effect (SPGE), also called surface photovoltaic effect[22], refers to the photovoltaic (PV) effect generated on the surface of materials. SPGE can be induced by asymmetric scattering of the surface within an oblique incidence[22] or other distinctive surface features. The well-known circular photogalvanic effect (CPGE) is common on the surface of topological insulators, which is also a case of SPGE. Theory based on both shift and ballistic current suggests that the SPGE response, containing linear and circular components, is strongly linked to the surface state of topological insulators[23–25] and Weyl semimetals[26–28]. Experimental investigations have primarily focused on the abundant polarization characteristics and demonstrated SPGE as an efficient probe for topological surfaces[29–31]. Meanwhile, SPGE serves as an alternative approach for the PV to be excited within the boundary-confined surface. SPGE-based photovoltaic devices offer unique advantages in terms of naturally-reduced symmetry. Here, we observe an intrinsic photocurrent response generated on the surface of the centrosymmetric material $Ag_2Te$ and it exhibits an ultra broad operation range spanning from visible to mid-infrared wavelengths.

## Results

### Crystal structure and photocurrent response of $Ag_2Te$

Monoclinic $Ag_2Te$ ($\beta$-$Ag_2Te$) has been both theoretically predicted and experimentally confirmed as a three-dimensional topological insulator with a narrow-bandgap bulk state. This material exhibits exceptional transport and optical properties in previous studies, including high carrier mobility and strong anisotropy[32–37]. The crystal structure of $Ag_2Te$, as depicted in Fig. 1a–b, belongs to the space group $P2_1/c$ (No. 14) and the point group $C_{2h}$. There are three symmetry operations: twofold screw rotation symmetry with the axis $C_{2b}$ along the crystallographic $b$-axis, glide mirror symmetry with the plane $\tilde{M}_b$ perpendicular to the crystallographic $b$-axis, and inversion symmetry, which is the product of the previous two operations. As shown in Fig. 1b, the glide operations orientate along $b$-axis and $c$-axis in the screw rotation and the glide mirror, respectively. In this study, we employed $Ag_2Te$ nanoplates grown via chemical vapor deposition (CVD)[33]. These nanoplates generally have a parallelogram shape, where the $b$-axis is consistently in-plane and lies parallel or perpendicular to one of their edges.

Although $Ag_2Te$ is not expected to exhibit bulk photovoltaic effect (BPVE) due to its inversion symmetry, we observed distinct short-circuit photocurrent in a standard two-terminal $Ag_2Te$ device, as shown in the inset of Fig. 1c. A 690-nm laser was normally incident on the sample. The laser power dependence of the photocurrent, as displayed in Fig. 1c, was measured at the center of the device (marked by the red dot in the inset) and shows a near-linear tendency, well fitted by $I \propto P^{1.1}$. The power factor of 1.1 suggests a second-order nonlinear effect dominant in the photocurrent response that shares the same form as the traditional BPVE expressed in Eq. (1). The slightly superlinear behavior is probably induced by the heating of the electron gas[31]. Photocurrent mapping was performed to investigate this anomalous response. As shown in Fig. 1d, it can be excited anywhere within the $Ag_2Te$ sample, and its distribution is relatively uniform, with a magnitude of approximately 70 nA. Based on the mapping results, we can eliminate two probable mechanisms for the photocurrent response. Firstly, there are merely Schottky junctions located on the contacts that form built-in electric fields in the pristine $Ag_2Te$ device. Thus, the built-in electric field-induced PV certainly exists near the contacts and presents opposite signals on the two contacts. Secondly, the photo-thermoelectric effect (PTE), arising from laser-induced temperature gradient coupled with inhomogeneous or anisotropic Seebeck coefficient distribution[38–40], should display an in-plane odd-symmetric distribution due to the mirror-symmetric device structure, regardless of whether the abnormal Seebeck coefficient distribution arises from $Ag_2Te$ itself or the contacts. Therefore, the unidirectional and uniform photocurrent response in $Ag_2Te$ definitely takes another mechanism providing an asymmetric factor to break the mirror-symmetric device structure. Besides, a small 'displacement' between the photocurrent mapping and available $Ag_2Te$ area should be noticed, which is attributed to a mixture of Schottky junction-induced PV located near the contacts added to this unknown uniform photocurrent response. This phenomenon emerges in multiple devices and is confirmed in Supplementary Fig. 2.

### Symmetry analysis

To explore the underlying mechanism, the crystallographic orientation dependence of the photocurrent response was investigated, as shown in Fig. 2a–c. A rhomboid-shaped $Ag_2Te$ nanoplate outlined by dot lines was etched into four pieces and finally fabricated into five samples (labeled s1-s5) with different orientations for photocurrent collection (Fig. 2a). The blue regions represent the reserved $Ag_2Te$ after the etching process. With the two pairs of parallel edges of the initial $Ag_2Te$ nanoplate labeled as e1 and e2, the orientations of the five samples can be described as follows: s1 and s5 along e1, s2 perpendicular to e2, s3 along e2, and s4 perpendicular to e1. One of the electrodes in the center region was designed to connect all five samples at one end of each. This electrode was wired to a lock-in amplifier and other electrodes were grounded. This configuration allows us to record the responses of all the samples in a single photocurrent mapping, as shown in Fig. 2b. Note that the photocurrent in s1 appears to have the reverse sign compared to s5 in the mapping, but in reality, the directions of the photocurrent in these two samples are consistent due to the opposite grounded sides. The line profiles of photocurrent along the red arrows for the five samples obtained from Fig. 2b are displayed in Fig. 2c. The result clearly demonstrates an orientation dependence. The anomalous photocurrent response, characterized by a uniform distribution over the samples, is observed in s1, s2, s4, and s5. Among these, the responses in s1, s2, and s5 are significant, while the response in s4 is relatively weak, accompanied by strong PV induced by Schottky junctions near the contacts. In contrast, s3 exhibits no such anomalous response but PV induced by Schottky junctions. A preliminary conclusion can be drawn that along a specific crystallographic orientation, namely, along the edge e2 in this nanoplate, the anomalous photocurrent response is prohibited. Conversely,

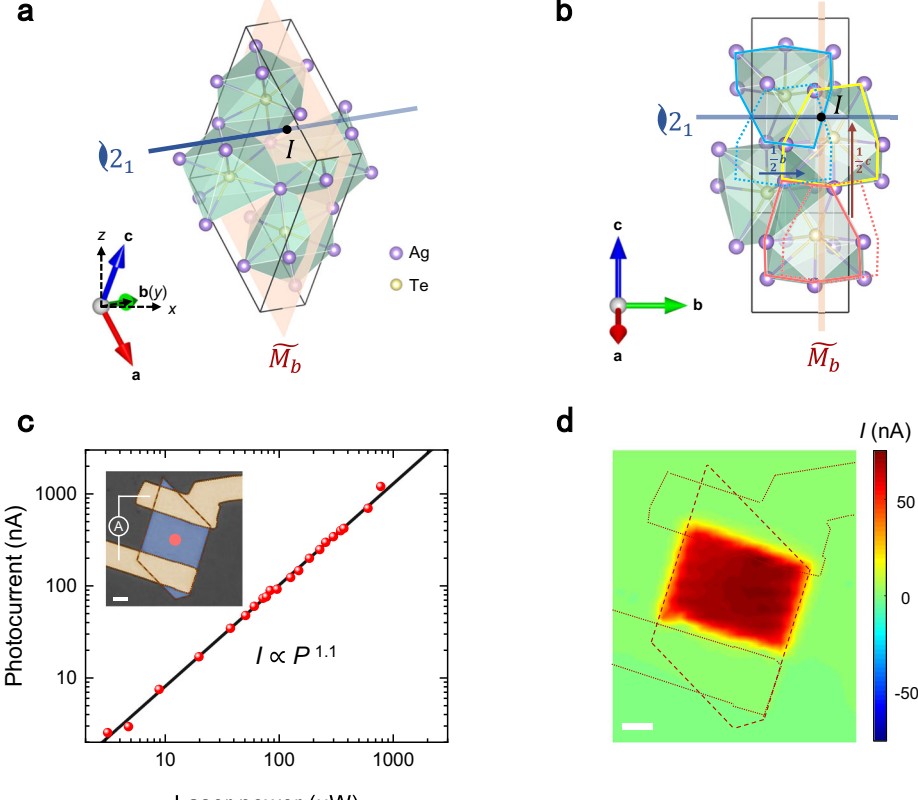

**Fig. 1 | Crystal structure and photocurrent response of Ag₂Te. a, b** Unit cell of Ag₂Te in three-dimensional perspective (**a**) and the projection of (100) plane (**b**). The inky blue axis, the orange-pink plane, and the black dot indicate the screw rotation axis $C_{2b}$, the glide mirror plane $\widetilde{M}_b$ and the center of inversion symmetry, respectively. The colorized axes show the unit cell vectors. The dashed black axes in **a** show the laboratory coordinates, where the nanoplate surface ($\bar{1}01$) is parallel to the *x-y* plane. **b** Schematic of the microscopic symmetry operations. The blue-outlined polyhedron transforms to the dotted blue one after the operation of a pure rotation and then coincides with the yellow one after a glide of 1/2*b* along the *b*-axis. The red-outlined polyhedron transforms to the dotted red one after the operation of a pure mirror and then coincides with the yellow one after a glide of 1/2*c* along the *c*-axis. **c** Main: power dependence of the photocurrent response in Ag₂Te. The red points are the measured data and the black solid line is the fitting curve of $I \propto P^{1.1}$, respectively. Inset: false-color optical image of the device with the scheme of the short-circuit photocurrent measurement. **d** Mapping of the photocurrent response of the device. The laser wavelength in **c**, **d** is 690 nm and the power in **d** is ~70 μW. The scale bars are all 5 μm.

the orientation perpendicular to e2 supports the anomalous photocurrent.

Based on the selectivity of crystallographic orientation and the second-order character indicated by the near-linear power dependence, we revisited Eq. (1) and speculated that the photocurrent response is generated on the surface of Ag₂Te as SPGE, where the symmetry is reduced due to the boundary nature. As shown in the aforementioned crystal structure (Fig. 1a–b), the *c*-axis is mostly out-of-plane and the *b*-axis is completely in-plane. Therefore, the translational symmetry along the *c*-axis is broken at the surface, leading to the breaking of glide mirror symmetry $\widetilde{M}_b$. In contrast, the glide along the *b*-axis is insusceptible so the screw rotation symmetry $C_{2b}$ is preserved at the surface. The symmetry analysis of the bulk and surface of Ag₂Te is provided in Table 1, revealing that the third-rank tensor $\beta$, which is absent in the bulk, reappears on the surface. Under unpolarized illumination, only the photocurrent along the *b*-axis is permitted owing to $\beta_{yxx}$ and $\beta_{yyy}$. It's consistent with the measurement, which also indicates *b*-axis is perpendicular to the edge e2. In Supplementary Note 8, we constructed a slab model to represent an isolated surface and theoretically calculated the shift current. As expected from symmetry analysis, shift current is activated along the *b*-axis of the slab while prohibited in the bulk structure.

The laser polarization dependence of the photocurrent is examined for further understanding. Figure 2d shows an expected result for $I_{\perp \mathbf{b}}$ and $I_{\parallel \mathbf{b}}$. The photocurrent response $I_{\parallel \mathbf{b}}$, determined by the elements $\beta_{yxx}$ and $\beta_{yyy}$, exhibits a twofold symmetric anisotropy with an anisotropy ratio determined by the quotient of the two, as shown in Fig. 2d for the case when $\beta_{yyy} = 0.5\beta_{yxx}$. On the other hand, the element $\beta_{xxy}$ and the corresponding photocurrent response $I_{\perp \mathbf{b}}$ signify purely bipolar anisotropy. Figure 2e displays the measured polarization dependence of s2, where the photocurrent flows perpendicular to the edge e2. Under illumination at 690 nm, the photocurrent is nearly isotropic. While at 1310 nm, it exhibits a twofold symmetric pattern and reaches a maximum with the laser polarized perpendicular to the edge e2. The difference between the two wavelengths can be attributed to variations of $\beta_{yxx}$ and $\beta_{yyy}$ over the spectrum. However, there is no observable signal in the center of s3 under illumination of any polarization and wavelength, implying a negligible $\beta_{xxy}$. We suggest that the orientation perpendicular to **b** possesses higher symmetry compared to the screw rotation axis **b**, making it more difficult to generate photocurrent. A similar phenomenon was observed in monolayer WSe₂[17], which was aforementioned regarding the symmetry constraint of BPVE. A more comprehensive discussion is provided in Supplementary Note 5.

Though the symmetry analysis and the theoretical calculation solely consider an isolated surface, the result can emerge in an overall crystal. For a sufficiently thick sample, only the top surface is involved. Whereas both top and bottom surfaces contribute when the light penetration depth is comparable to the thickness of the sample. The tensor $\beta$ of the two surfaces in a centrosymmetric material, such as

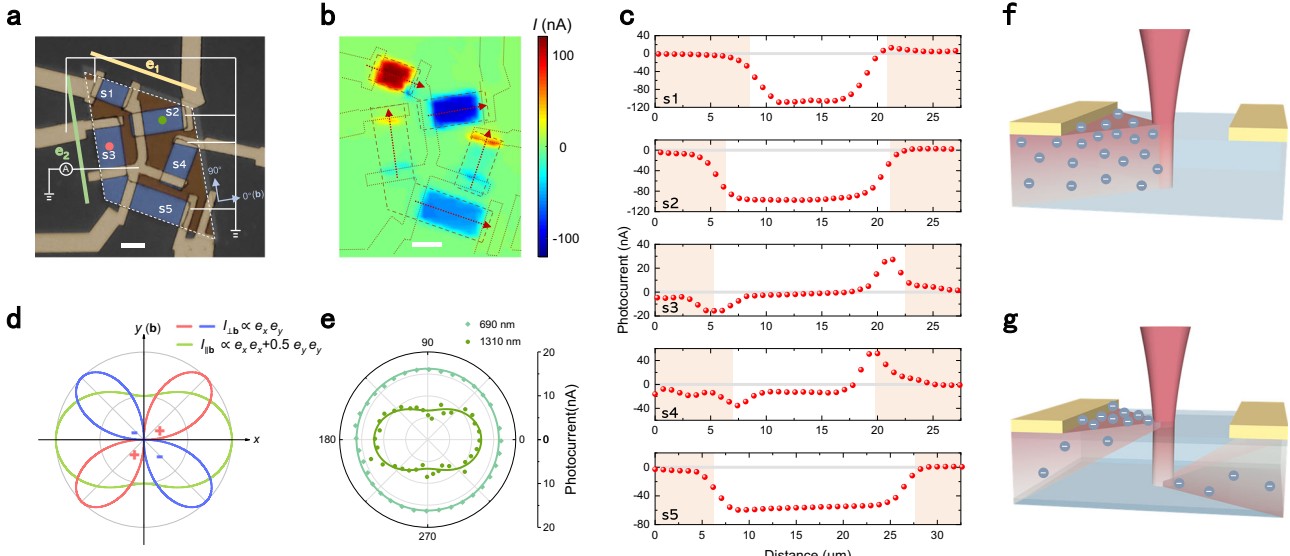

**Fig. 2 | Crystallographic orientation and linear polarization dependence of the photocurrent response in Ag₂Te. a** False-color optical image of an Ag₂Te nanoplate etched into 5 samples (s1-s5). The white dotted rhomboid outlines the initial Ag₂Te nanoplate. The yellow and green bars mark the two edges of the nanoplate, e1 and e2. **b** Photocurrent mapping measured with a wiring scheme depicted in **a** of the device. The laser wavelength is 690 nm and the power is ~100 μW. The scale bars in **a**, **b** are all 10 μm. **c** Line-profile of the photocurrent extracted from (**b**) along the red arrows marked in (**b**). The red arrows also indicate the photocurrent flow directions. The orange-pink shaded areas in **c** represent the electrode regions. **d** Simulated polarization dependence of $I_{\perp b}$ (photocurrent perpendicular to $b$-axis) and $I_{\parallel b}$ (photocurrent along $b$-axis). $e_x$ and $e_y$ are the light polarization unit vectors. **e** Polarization dependence measured at the center of s2 marked by the green dot in

**a** under the illumination of 690 nm and 1310 nm. The fitting curve of 690 nm is a constant corresponding to $\beta_{yyy} = \beta_{yxx}$. The fitting curve of 1310 nm is $I = 9.40 + 2.76\cos(2(\varphi - 0.109°))$, where $\varphi$ is the polarization angle, corresponding to $\beta_{yxx} = 0.546\beta_{yyy}$. The polarization dependence of s3 was also measured at its center marked by the red dot in **a**, but the signal always remains equal to zero. Light blue coordinates in **a** mark the directions of the polar coordinates. **f**, **g** Schematics of the bulk photovoltaic effect (BPVE) (**f**) and the surface photogalvanic effect (SPGE) (**g**). In BPVE, a unidirectional photocurrent is generated in the bulk of non-centrosymmetric materials under illumination. In SPGE, photocurrent flows are generated on the top and bottom surfaces, allowed by specific surface symmetry, and often produce a net photocurrent due to the bulk absorption.

Ag₂Te, should be odd symmetric and cancel each other out in a uniform electric field. Nevertheless, in practice, the intensity of illumination on the top surface is stronger than on the bottom surface due to bulk absorption, resulting in a net photocurrent. Furthermore, for materials with asymmetric top and bottom surfaces, induced by intrinsic structure or environment, the tensor $\beta$ could be irrelevant and even stimulate available photocurrent in extremely thin samples. The case in Ag₂Te is investigated in the later. Generally, the operating processes of BPVE and SPGE are illustrated in Fig. 2f, g. In traditional BPVE, a unidirectional photocurrent is generated throughout the bulk of non-centrosymmetric materials. In SPGE, two photocurrent flows are respectively generated from the top and bottom surface and produce a net current, which is often along the direction of the top surface photocurrent. While a non-centrosymmetric crystal structure is not necessary for SPGE, a specific low-symmetric surface that yields a nonvanishing $\beta$ is still requested.

**Table 1 | Symmetry operations and corresponding non-vanishing elements of third-rank tensor $\beta$ in the bulk and surface of Ag₂Te**

| | | Bulk | Surface (//b) |
|---|---|---|---|
| Symmetry operations | $\widetilde{M_b}$ | √ | × |
| | $C_{2b}$ | √ | √ |
| | $I$ | √ | × |
| Space group | | $P2_1/c$ | $P2_1$ |
| Nonvanishing elements of $\beta$ | | none | $\beta_{yxx}, \beta_{yyy}, \beta_{xxy}$ |

Note that only elements leading to in-plane photocurrent with normal incidence are listed here. The subscripts $x$, $y$, and $z$ are Cartesian coordinates indicated in Fig. 1a where $y$ direction is parallel to the crystallographic $b$-axis.

## Elimination of external effects

To eliminate the potential influence of the environment, specifically the Ag₂Te-substrate interface and Ag₂Te-vacuum boundary, we fabricated a sandwich-like device. As depicted in Fig. 3a-c, this structure consists of an Ag₂Te nanoplate placed entirely on the bottom hexagonal boron nitride (hBN) and partially covered by the top-hBN. The photocurrent mapping (refer to Fig. 3c) of the hBN-Ag₂Te-hBN and hBN-Ag₂Te-vacuum regions reveals nearly indistinguishable responses, uniformly distributed over the nanoplate. We should note that the extinction coefficient of hBN is approximately zero[41], so the illumination intensity on the regions uncovered and covered by top-hBN (with a thickness of ~40 nm) is nearly the same. As a result, their equivalent photocurrent responses are expected, despite color differences observed in the optical image due to variations in reflection. Then, since we obtained consistent results within three different environments (SiO₂-Ag₂Te-vacuum, hBN-Ag₂Te-vacuum, and hBN-Ag₂Te-hBN), we can exclude potential extrinsic effects such as structural asymmetry induced by the different boundaries of the top and bottom surfaces, and strain induced by the substrate-Ag₂Te interface. In fact, strain cannot be the dominant source of the observed photocurrent in principle, due to the inversion symmetry robust against strain. A detailed discussion about the elimination of strain is provided in Supplementary Note 6. Otherwise, SPGE as an intrinsic characteristic accounts for the photocurrent response in Ag₂Te.

## The surface origin

To further identify the surface origin of the photocurrent, a turn-over experiment is carried out (Fig. 3d–k). One piece of Ag₂Te nanoplate with side A up on a polydimethylsiloxane (PDMS) stamp was mechanically cut into fragments, as shown in Fig. 3e. Subsequently, some of the fragments were lifted up and turned over to have side B up

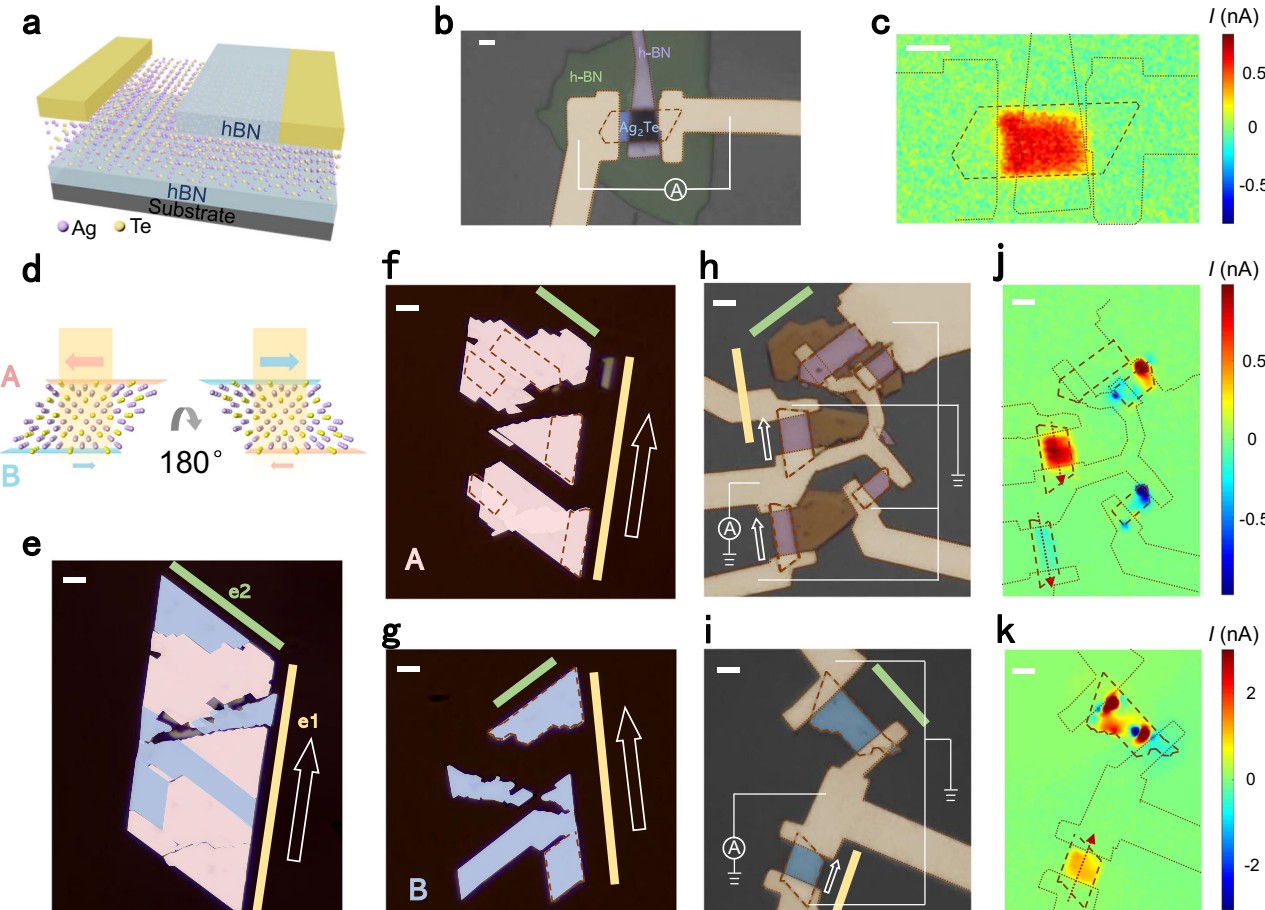

**Fig. 3 | Evidence for the surface origin of the photocurrent response in Ag₂Te.**
**a–c**, Schematic structure of the hBN(hexagonal born nitride)-Ag₂Te-hBN device (**a**), its false-color optical image with scheme of the measurement (**b**), and photo-current response mapping (**c**). Outlines of Ag₂Te, electrodes, and top-hBN have been marked with dotted lines. **d–k**, The turn-over experiment of Ag₂Te. **d** Schematic of the turn-over process. The pink and blue arrows represent the opposite responses generated from surfaces A and B, respectively. The direction of the total response is determined by the upper surface, which would be reversed if the sample is turned over. **e–g** False-color optical images of the sample. One Ag₂Te nanoplate on polydimethylsiloxane (PDMS) (**e**) was mechanically cut into

fragments and parts of them (recolored as blue) were lifted up and consequently turned over to B side up (**g**) by another PDMS. The rest parts (recolored as pink) keep A side up (**f**) the same as the initial nanoplate shown in (**e**). **h–k**, False-color optical images and photocurrent mapping results of the two devices fabricated from (**f**) and (**g**), respectively. The red dashed lines in **f–k** outlines the samples through the fabrication procedures. The yellow and green bars in **e–i** mark the two edges of the nanoplate, e1 and e2. White arrows in **e–i** indicate the same crystal-lographic direction along e1. Red arrows in **j**, **k** indicate the photocurrent direc-tions. The laser wavelength is 690 nm and the power is ~80 μW in **c**, **j**, **k**. The scale bars are all 5 μm.

using another PDMS stamp, as displayed in Fig. 3g, while the remaining fragments retained side A up (Fig. 3f) as same as the initial nanoplate. These two sets of fragments were then fabricated into devices depic-ted in Fig. 3h, i, respectively, with opposite sides up. A series of etching procedures, electrode distribution, and wiring configuration, similar to Fig. 2a, were implemented in the devices in Fig. 3h, i to determine their crystallographic orientations. The topper sample in Fig. 3h is dis-connected due to the dropped electrode on its left. As shown in Fig. 3j, k, all the other samples along the edge e2 exhibit complex PV and PTE responses. On the contrary, the samples along the edge e1 show a uniform photocurrent response. Therefore the *b*-axis is recognized perpendicular to the edge e2. The directions of the pho-tocurrent along e1 are indicated by the red arrows in Fig. 3j, k. Mean-while, a vector parallel to e1 was marked, and its direction was carefully tracked throughout the fabrication processes, as depicted by the white hollow arrows in Fig. 3e–i. By comparing the two sets of arrows in Fig. 3h–k, it was observed that the photocurrent directions of the two devices with opposite sides up were also opposite referring to the crystallographic orientation. This result definitely confirms the sur-face origin of the photocurrent since a response generated in the bulk would maintain its direction regardless of the turn-over operation,

whereas a response generated from the opposite surfaces could change sign depending on the upper surface, as illustrated in Fig. 3d.

Results of the sandwich-like device and the turn-over experiment suggest that the SPGE in Ag₂Te doesn't rely on asymmetric factors in the environment and the two surfaces naturally host opposite photo-current flows, hence the illumination loss during the penetration is the decisive factor for the net photocurrent. A thickness-dependent measurement was carried out to verify it. As shown in Fig. 4a, thicker samples result in larger responsivity, since the laser gradually becomes incapable of penetrating the bottom surface of Ag₂Te, thereby effec-tively suppressing the counteraction of the two opposite responses. The data is fitted by:

$$R = R_0(1 - \exp(-\alpha d)) \qquad (2)$$

where $R_0$ is the responsivity of an individual surface, $d$ is the thick-ness of Ag₂Te and $\alpha$ is the transmission coefficient with a magnitude of $3.4 \times 10^6$ m⁻¹ measured under a 690-nm illustration used in this responsivity measurement. The fitted $R_0$ is 0.17 nA/(Wcm⁻²) and the measured largest responsivity of 0.14 nA/(Wcm⁻²) is obtained from the thickest sample (~646 nm). The well-fitted result substantiates

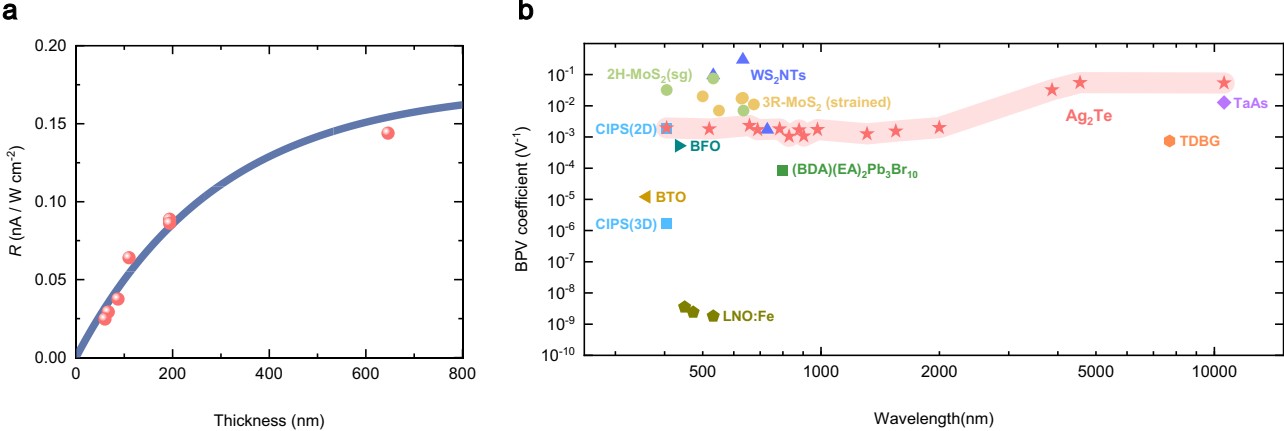

**Fig. 4 | Performance of the surface photogalvanic effect in Ag$_2$Te. a** Thickness dependence of the responsivity of Ag$_2$Te. Measurements of all the samples are carried out under the same 690-nm illumination with a power of ~80μW. The data is fitted by $R = R_0(1 - \exp(-\alpha d))$, where $R_0$ is the responsivity of an individual surface, $d$ is the thickness of Ag$_2$Te and $\alpha$ is the transmission coefficient. **b** Measured BPV coefficients of non-centrosymmetric materials (WS$_2$ nanotube[17], 2H-MoS$_2$ with strain gradient (sg)[19], strained 3R-MoS$_2$[13], CuInP$_2$S$_6$ (CIPS) (2D and 3D)[44], BaTiO$_3$ (BTO)[45], BiFeO$_3$ (BFO)[46], (BDA)(EA)$_2$Pb$_3$Br$_{10}$[47], Fe-doped LiNbO$_3$ (LNO:Fe)[48], twisted double bilayer graphene (TDBG)[21], TaAs[15]) and equivalent PV coefficient of Ag$_2$Te. Data of (BDA)(EA)$_2$Pb$_3$Br$_{10}$ and TDBG are taken from their responsivity, data of LNO:Fe is $\beta_{333}$, and others are effective values of $\beta$.

the bulk absorption accounting for the net response generated from the two opposite surfaces. Therefore, SPGE should be excited in Ag$_2$Te with sufficient thickness. On the other hand, the requirement of thickness is unnecessary in materials with asymmetric top and bottom surfaces. For instance, misaligned Fermi arcs of the two surfaces due to different boundary conditions in Weyl semimetals are expected to produce net photocurrent even within a thin sample[26].

### Performance of SPGE in Ag$_2$Te

Finally, the performance of SPGE in Ag$_2$Te over a spectral range from visible to mid-infrared is evaluated and compared with other non-centrosymmetric BPVE materials. Although the photocurrent in Ag$_2$Te is generated on the surfaces, we calculated an equivalent PV coefficient for the purpose of comparison by considering the entire thickness of the nanoplate to estimate the photocurrent density. The data is attained from the thickest sample recorded in Fig. 4a and measured using a focused laser. Since the spot size of the laser is unable to fully cover the sample, there is a loss during the diffusion process. Consequently, the measured equivalent PV coefficient in Ag$_2$Te is on the low side. On the other hand, the performance in the mid-infrared range is significantly improved due to a larger spot size. Nevertheless, the performance of Ag$_2$Te remains relatively stable across the spectral range and demonstrates a significant advantage for mid-infrared applications, with an equivalent PV coefficient of approximately 0.054 V$^{-1}$ under 10.6-μm illumination. In contrast, most traditional BPVE materials can only operate with visible light, and so far, only a few emerging materials, such as gapless Weyl semimetal TaAs and narrow-bandgap twisted double bilayer graphene (TDBG), have been able to work in the mid-infrared range.

### Discussion

We note that the bandgap of Ag$_2$Te is estimated to be around 30 ~ 88 meV, as determined theoretically and experimentally[32,34,42,43]. However, this value is certainly lower than the minimum exciting energy used in our measurement, which is ~117 meV (10.6 μm). Up to now, there is no straightforward evidence of the involvement of the gapless Dirac cone in the topological surface state. To further investigate this, it would be beneficial to conduct experiments using lower excitation energies in proximity to the Dirac cone and circularly polarized illumination referring to spin-momentum locking[29], in order

to better understand the participation and achieve possible enhancement of the surface state.

In conclusion, this study demonstrates the presence of a surface photocurrent response in Ag$_2$Te, which is confined by the surface symmetry. Our experimental results confirm that the SPGE in Ag$_2$Te exhibits a defined crystallographic orientation selectivity and occurs as the difference between two opposite photocurrent flows generated from the top and bottom surfaces, which are unequal due to bulk absorption. Importantly, SPGE in Ag$_2$Te is activated across an ultra broad spectral range from visible to mid-infrared light, showcasing potential in solar cells and mid-infrared detector engineering. These results enrich our understanding of the PV and overcome the non-centrosymmetric constraints in BPVE, thereby expanding the range of potential materials for self-powered optoelectronic devices. In Ag$_2$Te, the symmetry reduction is associated with the out-of-plane glide. To explore SPGE in other systems, utilizing a high Miller index surface presents a promising avenue.

## Methods
### Crystal growth

Polycrystalline Ag$_2$Te powders were synthesized by reacting high-purity Ag and Te elements (99.999%, Alfa Aesar) at a reaction temperature of 1050 °C. Then Ag$_2$Te nanoplates were grown by chemical vapor deposition (CVD) using polycrystalline Ag$_2$Te powders as precursor and argon as carrier gas. The Ag$_2$Te powders were placed at the center of a horizontal tube furnace heated to 980–1050 °C, and Ag$_2$Te nanoplates were finally deposited on *c*-cut sapphire substrates put downstream slantways[33].

### Device fabrication

The CVD-grown Ag$_2$Te nanoplates were dry-transferred onto a Si substrate capped with a 285nm-thick SiO$_2$ layer by poly-dimethylsiloxane (PDMS) stamps. The etched devices were then additionally patterned by a standard electron-beam lithography technique and etched into pieces by ion beam etching (IBE) with argon for ~4 min. As for the hBN-Ag$_2$Te-hBN samples, hBN flakes were first mechanically exfoliated onto PDMS stamps and transferred onto the Si/SiO$_2$ substrates. Then Ag$_2$Te and top-hBN were transferred onto the bottom-hBN successively. The turn-over samples were prepared following the steps below with detailed pictures shown in Supplementary Fig. 4: First, one piece of Ag$_2$Te nanoplate was transferred onto PDMS

stamp-1. Second, the tip of a probe was used to cut the nanoplate into two pieces. Third, PDMS stamp-2 was pressed on stamp-1 (for the sample in Fig. 3e–k, it was mechanically crushed into fragments during this step) and then released slowly, getting part of the Ag$_2$Te fragments transferred onto stamp-2. Finally, Ag$_2$Te fragments on PDMS stamp-1 and stamp-2 were all transferred onto Si/SiO$_2$ substrates. All the transfer, exfoliation, and cut-off works were carried out in a glove box. Finally, all the electrodes were patterned by electron-beam lithography and deposited with Cr (5 nm) /Au (100 nm) via magnetron sputtering.

## Characterizations

The laser sources used in the experiment are as follows: 405 nm, 520 nm, 658 nm, 690 nm, 785 nm, 830 nm, 880 nm, 904 nm, 980 nm, 1310 nm, and 1550 nm are a series of laser diodes from Thorlabs. The laser source of 2000nm is a fiber laser from NPI Lasers. The laser sources of 3870 nm and 4560 nm are two quantum cascade lasers from Thorlabs. Lastly, the laser source of 10600 nm is a CO$_2$ laser from ACCESS LASER. The laser beam is normal incident and focuses on the sample through an x40 objective (Olympus) for 405-2000nm and an x15 reflective objective (Thorlabs) for 3870 nm, 4560 nm, and 10600 nm. The sizes of focused laser spots of different wavelengths (405 ~ 10600 nm) are about 1.9, 1.8, 2.4, 2.1, 2.3, 1.8, 2.3, 2.0, 2.7, 2.5, 2.8, 3.8, 26.2, 28.7 and 54.4 μm, respectively. The sample was put in a vacuum chamber (< 0.1 mbar, room temperature) during the measurement, and the photocurrent mapping was performed by using a scanning galvo. The laser beam was modulated by a chopper of 37 Hz and the photocurrent and reflection intensity were measured simultaneously using two lock-in amplifiers (SR860). A linear polarizer and a half-wave plate (Thorlabs) were used in the polarization-dependent measurement. A source meter (Keithley 2400) was used in the I-V measurement and dc bias-dependent photovoltage measurement (Supplementary Fig. 1 and 2). The thickness of Ag$_2$Te samples was measured by a tapping mode atomic force microscope. The transmission coefficient was obtained from practical measured transmitted intensity of Ag$_2$Te nanoplates under the 690-nm illustration by light intensity meter.

## Data availability

The Source Data underlying the figures of this study are available with the paper. All raw data generated during the current study are available from the corresponding authors upon request. Source data are provided with this paper.

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

## Acknowledgements

F. X. was supported by the National Natural Science Foundation of China (52225207, 11934005, and 52350001), the Shanghai Pilot Program for Basic Research - FuDan University 21TQ1400100 (21TQ006), and the Shanghai Municipal Science and Technology Major Project (Grant No.2019SHZDZX01). J. Y. was supported by the National Natural Science Foundation of China (52222202). Y. G. is supported by the National Natural Science Foundation of China (Grant No.12374164) and the Innovation Program for Quantum Science and Technology (Grant No. 2021ZD0302802). W. Z. was supported by the National Key Research and Development Program of China (2019YFA0210004), the Innovation Program for Quantum Science and Technology (Grant No. 2021ZD0302800), the National Natural Science Foundation of China (Grant No. 12334004). We acknowledge D. Sun and X. Gan for their helpful discussions.

## Author contributions

F.X. conceived the ideas and supervised the overall research. P.L. synthesized high-quality Ag$_2$Te samples. X.X. fabricated the nanodevices with the help of Z.L., L.A., X.C, Z.J., P.L., Y.Z., and M.Z.. X.X. and J.Z. performed the photocurrent measurements with the help of P.L.. Z.D. and Y.G. performed the first-principles calculations. J.-S. Y., J.-Y. Y. and S. D. provided high-quality boron nitride samples and helped with some material characterization. X.X., P.L., Z.D., W.Z., Y.G., and F.X. analyzed and interpreted the results. X.X. and F.X. wrote the paper with assistance from all other co-authors.

## Competing interests

The authors declare no competing interests.
