## [Peer Review File · Nature Communications]

Surface photogalvanic effect in Ag_2TeEditorial Note: Parts of this Peer Review File have been redacted as indicated to remove third-party material where no permission to publish could be obtained.

REVIEWER COMMENTS

Reviewer #1 (Remarks to the Author):

Xiaoyi Xie et al. report a photocurrent response in Ag₂Te without an external electrical bias and attribute this observation to the surface photogalvanic effect in the centrosymmetric crystal.

The observation itself is intriguing; however, the authors need more evidence to support their claim.

Ag₂Te has a small band gap, and the I-V characteristic also shows that the device is highly conductive. Therefore, it is challenging to distinguish surface properties from bulk properties. Currently, it is not conclusive why the photocurrent is specifically generated from the surface properties.

The authors use the centrosymmetry of the crystal to argue against the contribution from the bulk crystal. However, there are many examples where crystals grow with intrinsic strain. Could the authors verify whether the effect of strain in the Ag₂Te bulk crystal is negligible?

The observed photocurrent spans a wide range of wavelengths and shows weak polarization dependence. Could the authors theoretically or numerically explain these observations rather than just showing symmetry analysis? How is this weak polarization dependence across wide wavelengths related to the surface property of Ag₂Te?

How do temperature and laser modulation frequency affect the observed photocurrent? Exploring these properties may help reveal the origin of the photocurrent in this study.

In conclusion, while I am intrigued by the phenomenon described, further work is necessary to substantiate the authors' claims.

Reviewer #2 (Remarks to the Author):

Photovoltaic effect reflecting the symmetry breaking of crystals is now one of the central topics in condensed matter physics due to its potential of overcoming the Shockley-Queisser limit in the conventional solar cells and its intrinsic mechanisms related with the carrier dynamics or band geometry/topology. Generally, non-centrosymmetric crystal structures is required to observe such intrinsic photovoltaic effect.

In this paper, authors reported the surface photogalvanic effect in centrosymmetric Ag₂Te. Although bulk crystal has a centrosymmetric crystal structure, inversion symmetry is broken at the surface, causing large photovoltaic response. They demonstrated that top and bottom surfaces generate the photocurrent of opposite sign and also clarified that photovoltaic response is

activated across an ultrabroad spectral range from visible to mid-infrared light.

I think these findings provide a new direction of studying the photovoltaic effect and useful for applications in solar cells.

I have several comments and questions below.

1. It is difficult to intuitively understand how symmetry is reduced at the surface of Ag₂Te. Is there in-plane polarization on the surface? I think it may be good to draw polyhedra and bonds made up of re-adjacent atoms in the schematics (Fig. 1 a, for example).
2. I think surface photogalvanic effect can in principle occur in many centrosymmetric materials but there have been a few reports so far. What is special for Ag₂Te? Does the topological surface state (high mobility carriers only near the surface or quantum geometric nature of the surface carriers) play the important role?
3. Related with the above question, I would like to know the authors' opinion on the possible origin of the observed surface photogalvanic effect. Do authors think that something similar to the bulk photovoltaic effect (shift current, ballistic current, injection current etc.) are happening on surfaces?
4. Is the bulk part completely unrelated to this phenomenon? If the carrier number changes in the bulk part, will it affect the surface photogalvanic effect?

Response to Reviewer's Comments

We acknowledge the reviewers for carefully reading our manuscript “Surface photogalvanic effect in Ag_2Te ” (NCOMMS-23-58651-T) and for their insightful comments which help us improve the quality of our work. According to their kind advice, we have carefully reviewed and revised the presentation of our manuscript. The corresponding revisions in response to their comments have been made. The revised parts have been highlighted in the updated manuscript. The detailed revisions are listed on a separate page at the end of this response letter.

Response to Reviewer #1

General comment:

Xiaoyi Xie et al. report a photocurrent response in Ag_2Te without an external electrical bias and attribute this observation to the surface photogalvanic effect in the centrosymmetric crystal.

The observation itself is intriguing; however, the authors need more evidence to support their claim.

Response:

We thank the reviewer for his/her helpful comments on our manuscript and we appreciate that the reviewer finds the topic interesting. We especially appreciate the specific questions and suggestions, which help us to further improve our manuscript.

In the following, we would like to address all the comments point-by-point and highlight the modifications made to the manuscript. We hope that our revised manuscript can remove the reviewer's concerns.

Comment 1:

Ag_2Te has a small band gap, and the I - V characteristic also shows that the device is highly conductive. Therefore, it is challenging to distinguish surface properties from bulk properties. Currently, it is not conclusive why the photocurrent is specifically generated from the surface properties.

The authors use the centrosymmetry of the crystal to argue against the contribution from the bulk crystal. However, there are many examples where crystals grow with intrinsic strain. Could the authors verify whether the effect of strain in the Ag_2Te bulk crystal is negligible?

Response:

We agree with the reviewer that it's challenging to distinguish surface properties from bulk properties in this system and the concern about strain is justified. In the original manuscript, the turn-over experiment and substrate replacement experiment were carried out to confirm the surface origin and to eliminate the effect of strain, respectively. In the following, we will discuss the issue more comprehensively in

several aspects.

1) **Apart from the centrosymmetry of the crystal, the turn-over experiment is a significant evidence of the surface origin in our claim.** Though the bulk is highly conductive, which makes it difficult to distinguish surface and bulk signals electrically, the two can be identified by means of the photocurrent flow direction.

The photocurrent flow direction of the bulk photovoltaic effect (BPVE) depends on the polarity of the crystal. For instance, we assume BPVE is active along the x-axis in Ag_2Te (see the green arrow in Fig. R1a). The signal would always hold its direction after any rotation operations along this axis, such as C_{2x} . On the other hand, signals generated from the top surface would be reversed due to the altered effective surface (Fig.R1b). Note that the selection of the rotation axis doesn't influence the conclusion once we take the crystallographic orientation as a reference.

Experimentally, we cut off one piece of Ag_2Te nanoplate into two parts (colored pink and blue in Fig. R1c-e) and turned one of the parts over. The white arrows indicate the same crystallographic direction and the red arrows indicate the measured photocurrent directions (Fig. R1d,e). The two parts with opposite top surfaces were found to produce opposite flows, which proved that the photocurrent was generated from the surface, instead of the bulk.

Figure R1 | The turn-over experiment. (b-e are copied from Fig.3 in the main text. More experiment details are presented in Fig.3 and Fig.S4 in the supplementary information.)

2) Strain is a common factor in activating and/or enhancing BPVE, which has been reported in several studies. **However, strain is not the dominant source of the photocurrent in Ag_2Te .** Aside from the analysis of photocurrent flow direction in the turn-over experiment, we would focus on the case of strain in Ag_2Te in the following and verify it.

In principle, strain could break both rotation and mirror symmetry. Most of the

relevant works involve in-plane rotation symmetry, such as the bulk piezo-photovoltaic effect which was observed in 3R-MoS₂ enhancing its BPVE for the broken C_{3z} by artificially constructed strain¹. Unexpected circular BPVE is observed in twisted double bilayer graphene for the broken C_{3z} by possible strain². BPVE is observed in twisted bilayer graphene because the point group D_6 is reduced to C_1 by strain due to lattice reconstruction³. In practice, the C_{3z} and C_{2z} symmetries in twisted bilayer graphene systems are commonly broken by the strain and the encapsulation with hexagonal boron nitride (h-BN)⁴. **Nevertheless, we should emphasize that strain is helpless to break inversion symmetry.** As shown in Fig. R2, a uniaxial strain along the appropriate orientation will break the screw rotation symmetry C_{2b} and the glide mirror symmetry \tilde{M}_b in Ag₂Te, while the inversion symmetry is stable.

Figure R2 | Schematic of the symmetry of Ag₂Te (top view) without (a) and with (b) a uniaxial strain.

However, several randomly distributed strains could certainly break the inversion symmetry. The distortion within the Ag₂Te nanoplatform would be intricate in this case. Therefore, the photocurrent distribution (both sign and magnitude) would be also intricate and differ from our results which are highly uniform as shown in the photocurrent mapping (for example, Fig. 1d in the main text).

In addition, another possible approach to breaking the inversion symmetry is the strain gradient, referred to flexo-photovoltaic effect, which could be introduced by stacking or AFM tip^{5,6}. However, our mapping results also confirm that the strain gradient should be uniform. A uniformly distributed strain gradient covering the entire Ag₂Te nanoplatform is almost impossible to emerge spontaneously in numerous samples. Therefore, we declare that strain or strain gradient is hard to account for the photocurrent in Ag₂Te in principle. Meanwhile, we can carefully verify it through a few experiments.

Possible strain/strain gradient can be introduced during the procedure of crystal growth, transfer, lattice alignment with substrate, or particular design. Among these procedures, the crystal growth is not likely to introduce strain in our experiments. We used chemical vapor deposition (CVD) to grow Ag₂Te nanoplates where the nanoplates were deposited on some small grains, rather than adhered to the substrate. The details of the CVD growth can be seen in our previous work⁷. However, we would focus on the effect of strain in the following but not thoroughly investigate its probable source.

2.1) A previous work of our collaborators applied terahertz nanoscopy to image in-plane anisotropic plasmon polaritons in Ag_2Te grown by us⁸. In the scattering-type scanning near-field optical microscopy (s-SNOM) image shown in Fig.R3e, explicit interference fringes signify the propagating of polaritons which were reflected at the nanoplate edges. **The high-quality imaging of the interference fringes illustrates that there is no distinct local strain at the interior of the nanoplate.** Otherwise, the interference fringes would be interrupted and the strain can be detected in the s-SNOM image⁹.

[REDACTED]

Figure R3 | (a-c) Terahertz Near-field imaging of Ag_2Te (copied from reference⁸). (a) Schematic of the s-SNOM experiment. (b,c) Topography image (b) and THz near-field image (c) of part of one Ag_2Te nanoplate.

2.2) In the main text, we explored the photocurrent response within different environments. The sandwich structure (shown in Fig. R4) was designed to achieve the comparison of hBN- Ag_2Te -hBN, hBN- Ag_2Te -vacuum, and SiO_2 - Ag_2Te -vacuum. Characteristics of the responses don't show evident differences in the three cases. **Therefore, possible strain induced by lattice alignment with the substrate can be excluded.**

Figure R4 | Optical image (a) and photocurrent mapping (b) of the hBN- Ag_2Te -hBN device. (copied from Fig. 3 in the main text)

2.3) We artificially applied strain on the nanoplate utilizing a groove structure (referring to the 3R-MoS₂ work¹, shown in Fig. R5a,b) to probe the effect of uniaxial strain. Fig. R5c-f demonstrates our results. The SiO_2/Si substrate was pre-patterned by reactive ion etching (RIE) to form a groove with a depth of ~ 95 nm, outlined by the green dashed rectangle in Fig. R5c,e. The Ag_2Te nanoplate was dry-transferred onto the substrate and pushed down to the bottom of the groove by polydimethylsiloxane (PDMS). The altitude of Ag_2Te stepped over the groove was measured to confirm that the nanoplate was completely pushed down, as shown in Fig. R5d. Uniformly distributed strain would be applied in the center region of the groove.

The photocurrent mapping is shown in Fig. R5e-f, which is rather uniform over the whole nanoplate. **There is no evident difference within and out of the groove**

region, suggesting there is no evident enhancement to the photocurrent in Ag₂Te by uniaxial strain. The result is expectable, as we discussed above that strain cannot break the inversion symmetry.

Supposing that the photocurrent in Ag₂Te is induced by some kind of unknown strain, the appropriate orientation of the strain should be consistent with the photocurrent. Accordingly, the geometry (relative orientation between the groove and Ag₂Te) of the device shown in Fig. R5c-f is sufficient to check. **Now that the artificially applied strain cannot adjust the photocurrent in Ag₂Te, the photocurrent itself is unlikely induced by strain either.**

[REDACTED]

Figure R5 | The effect of artificially applied strain. (a,b) Schematic of the strain structure in the 3R-MoS₂ work, copied from reference¹. (c-f) Our experiment of Ag₂Te. (c,e) Optical image (c) and photocurrent mapping (e) of the strain device. The inset of (e) is the reflection image. (d) Depth of the groove measured by atomic force microscopy (AFM). The two lines are marked in (c) colored as black and red, respectively. (f) The photocurrent distribution along the white arrow indicated in (c). The laser wavelength is 690nm and the power is ~7μW.

2.4) We observed a few Ag₂Te nanoplates comprised of two grains with distinguished

crystallographic arrangements, possibly owing to perturbation during the CVD growth. The grain boundary (GB) has the potential to be formed with complicated strain /strain gradient /distortion /symmetry reduction. Activation and enhancement of BPVE have been observed at the GB in ReS₂¹⁰ and BFO¹¹, respectively. We investigated photocurrent distribution within such Ag₂Te nanoplates.

As shown in Fig. R6a-b, two trapezoidal grains are discovered under a polarized laser. The photocurrent mapping (Fig. R6c-d) illustrates the two grains with distinct crystallographic orientations, leading to different photocurrent directions and magnitudes, while there is no response near the GB. **The GB is just an interim of the two grains and the contribution of the possible strain/ strain gradient is negligible.**

Figure R6 | Ag₂Te nanoplate with a grain boundary. (a-c) Optical image with unpolarized white light (a), reflection mapping with polarized laser (b) and photocurrent mapping (c) of the device. The contrast ratio of the reflection image (b) is extremely adjusted to illustrate the two grains. (d) The photocurrent distribution along the red arrow is indicated in (c). The laser wavelength is 690 nm and the power is ~70 μW.

In a brief summary, the photocurrent response in Ag₂Te is indeed significant and resilient to additional applied strain and strain gradients. Both the potential spontaneous strain and its impact on the photocurrent is negligible, both theoretically and experimentally. We conclude that strain cannot be the primary source of the photocurrent. Instead, the reduction in symmetry at the surface explains the robust response.

Comment 2:

The observed photocurrent spans a wide range of wavelengths and shows weak polarization dependence. Could the authors theoretically or numerically explain these observations rather than just showing symmetry analysis? How is this weak polarization dependence across wide wavelengths related to the surface property of Ag₂Te?

Response:

The polarization dependence together with current orientation dependence reflects the symmetry of the BPV coefficient β_{qrs} . The former corresponds to r and s , and the latter corresponds to q . On one hand, our experiments (Fig. 2 of the main text) illustrate one distinct symmetry which confines the photocurrent orientation. Sorry that we mistook it as the glide mirror symmetry \widetilde{M}_y in the original manuscript and now we have corrected it as the screw rotation symmetry C_{2y} in the revised manuscript. Details are shown in response to the Comment 1 of Reviewer #2. On the other hand, there couldn't exist higher symmetry involving x and y (for example, in-plane rotation symmetry C_{4z}), which would lead to the vanishing of the photocurrent. **Therefore, there is no symmetry constraint on the relationship of β_{yxx} and β_{yyy} .**

1) **We measured the infrared absorption spectrum as a reference.** The absorption spectrum (reflection type) shown in Fig. R7 was measured with Ag₂Te nanoplate on the SiO₂/Si substrate using a Fourier transform infrared spectrometer (FTIR). The characteristic peak (~ 132 meV) and valley (~ 147 meV) come from the phonon of SiO₂ and carrier of doped Si, respectively. Though SiO₂ and Si are certainly isotropic, distinct two-fold symmetric polarization dependence was detected due to the coupling of Ag₂Te. **Apart from the characteristic peak and valley, the absorption curve shows aligned but weak polarization dependence over the infrared range. It is consistent with the polarization-dependent photocurrent as shown in Fig. S6 of the Supplementary Information.**

Note that the narrow bandgap of Ag₂Te exceeds the operation limit of FTIR. Consequently, we didn't observe the characteristic absorption peak of Ag₂Te itself. The main spectral range of our experiment, both photocurrent and absorption spectrum, involves multi-transitions far from the gap and the Dirac cone at the topological surface state. It's challenging to analyze the definite transition, especially in the visible and near-infrared range of our polarization-dependent photocurrent measurement. **However, the absorption spectrum provides experimental evidence that Ag₂Te has weak sensitivity to light polarization.**

Figure R7 | Polarization-dependent absorption spectrum of Ag₂Te. (a) The absorption spectrum under 0° and 90° polarized light. Inset: the optical image of the nanoplate with the crystallographic orientation. (b) Polarization dependence of the absorption at 132, 147, and 617 meV.

We should emphasize another issue that the main difference between the bulk and surface is the symmetry reduction. **However, the two symmetry operations in the bulk of Ag₂Te, C_{2y} and \widetilde{M}_y , give the same basic confinement to polarization dependence: two-fold symmetry along the high-symmetric axis b . The breaking of any of them wouldn't alter the confinement at the surface.** Therefore, we infer that the absorption tendency of polarization at the surface resembles the measured bulk absorption. As a result, we make the comparison above between the surface photocurrent and the bulk absorption. **In other words, it's hard to distinguish the surface property of the photocurrent from bulk if we merely analyze the polarization measurement.** Nevertheless, the main experimental evidence of the surface origin is the turn-over test (see Fig. 3 of the main text and the response to Comment 1).

2) The polarization characteristic of Ag₂Te is similar over different samples in that it's nearly isotropic under visible light and slightly two-fold symmetric under near-infrared light. However, the anisotropy ratio differs in various samples. The sample in Fig. 2 of the main text shows stronger anisotropy under 1310nm illumination than the one in Fig. S6 of the supplementary information. The practical measured results of the polarization dependence are affected by multiple factors. For example, apart from the absorption, anisotropic conductance also influences the measurement in different geometries. Variation of the band structure in different samples would influence the anisotropy of absorption as well. The measured polarization ratio I_{\max} / I_{\min} at 1310nm varies as 1.3 ~ 1.9 owing to the diversity of samples.

As an overview, BPVE in traditional ferroelectric materials^{12,13} and artificial microstructures^{14,15} with spectacular anisotropy generally show strong, or even bipolar polarization characteristics. On the contrary, BPVE in most non-ferroelectric materials shows weaker polarization dependence. Here we list typical polarization ratios of

several recent works in Table I. **Among these systems, the polarization ratio of Ag₂Te is at the normal level.**

Though strong polarization dependence was considered to be a key feature of BPVE decades ago, recent works in varied materials innovate this perspective. As we discussed above, proper symmetry constraints supporting the BPVE don't always restrict the light polarization dependence. **We attribute the strong polarization dependence to the effect of the strong anisotropy and electric polarization of the material itself.** The anisotropy degree could be checked in other optical phenomena, such as absorption. In Table I, the systems involving more anisotropic materials (BP and ReS₂) show relatively stronger polarization as well.

Moreover, the TDBG work² demonstrates that BPVE is sensitive to the band structure, which could be highly tuned by the double-gate in this system. **Their results show various polarization properties could emerge in BPVE according to the band structure.**

By the way, there is another case that the cross-item β_{qxy} (q is an arbitrary orientation of the current) would lead to a completely bipolar BPVE. However, this item is not observed in our work.

Table I | Polarization ratio of BPVE in non-ferroelectric materials.

Materials	I_{\max} / I_{\min}
WSe ₂ /BP ¹⁶	2
3R-MoS ₂ (s) ¹	1.3
2H-MoS ₂ (sg) ⁵	1.5
WS ₂ nanotube ¹⁷	1, 2 (different samples)
ReS ₂ GB ¹⁰	2
ReS ₂ (edge embedded) ¹⁸	-2.5, 2 (opposite edges)
TDBG ²	highly tunable in polarity, phase and ratio by double-gate
Ag ₂ Te (our work)	1 (690nm) / 1.3, 1.9 (1310nm, different samples)

3) **We performed first-principle calculations on the surface photogalvanic effect (SPGE) in Ag₂Te for deeper understanding.**

A slab model is constructed from the Ag₂Te crystal as shown in Fig. S1c, with both the top and bottom surfaces oriented along the $(\bar{1}01)$ plane. The process of slab extraction from the expanded unit cell is detailed in Fig. R8, showing the alignment of the slab with the original crystal axes. A vacuum layer has been added along the c -axis of the slab model to eliminate interactions from periodic boundaries in this direction. **We would use the slab model to represent an isolated surface structure in the following calculation.**

We performed electronic structure calculations using the Vienna Ab initio Simulation Package (VASP), adopting the Perdew-Burke-Ernzerhof (PBE) form of the generalized gradient approximation (GGA) for the exchange-correlation functional¹⁹⁻²¹. The kinetic energy cutoff for the plane-wave basis was set to 400 eV. For structural optimization and static self-consistent calculations, the Brillouin zone was sampled using Monkhorst-Pack k-point meshes of 9×17×1 and 15×27×1,

respectively. The positions of the atoms were fully optimized with a force convergence criterion of $-0.01\text{eV}/\text{\AA}$. To facilitate high-precision optical calculations, the energy convergence criterion for the electronic self-consistency was set to 10^{-7}eV . Additionally, a dipole correction was applied in the out-of-plane direction.

Figure R8 | Schematic of the slab model. (a) Unit cell of Ag_2Te . The blue plane ($\bar{1}01$) indicates the surface orientation of Ag_2Te nanoplate. (b) Side view of the bulk structure. (c) The slab model with a vacuum layer. The green dotted boxes in (b,c) show the extracted atom makeup of the slab.

We utilize the Shift Current (SC) framework to address issues related to the BPVE (SPGE), which mainly describes the interband absorption. Considering the electric field component of a monochromatic light, it can be expressed as:

$$\mathbf{E}(t) = \mathbf{E}(\omega)e^{-i\omega t} + \mathbf{E}(-\omega)e^{i\omega t}, \quad (1)$$

where \mathbf{E} represents the alternating current (AC) electric field of the light.

As a second-order optical response, the SC describes the direct current (DC) photocurrent output in the material's a-direction due to light polarized along the b-direction, expressed as:

$$J^a = \sigma_{bb}^a(\omega)\text{Re}[E_b(\omega)E_b(-\omega)], \quad (2)$$

where $\sigma_{bb}^a(\omega)$ represents the nonlinear optical conductivity induced by the SC, derived from Kubo formula. Note that σ_{bb}^a and the measured coefficient β_{abb} in the main text can be converted with each other according to $\beta_{abb} = 2\sigma_{bb}^a/(\epsilon_0 c)$, where ϵ_0 is the vacuum permittivity and c is the speed of light. $\sigma_{bb}^a(\omega)$ can be expressed as²²:

$$\sigma_{bb}^a(\omega) = -\frac{i\pi g_s e^3}{\hbar^2} \int [d\mathbf{k}] \sum_{nm} f_{nm} r_{mn}^b r_{nm}^{b;a} \times [\delta(\omega_{mn} - \omega) + \delta(\omega_{nm} - \omega)], \quad (3)$$

Here, g_s denotes the spin degeneracy, $f_{nm} = f_n - f_m$ and $\hbar\omega_{nm} = \varepsilon_n - \varepsilon_m$ are the differences in occupation numbers and energy eigenvalues between bands indexed by n and m at point \mathbf{k} , respectively (\mathbf{k} is omitted for simplicity). r_{mn}^b and $r_{nm}^{b;a}$ represent the interband dipole and its ‘generalized derivative’, which can be constructed from the Berry Connection $A_{knm}^a = i\langle u_{kn} | \partial_a u_{km} \rangle$, as:

$$\begin{aligned} r_{knm}^a &= (1 - \delta_{nm})A_{knm}^a, \\ r_{knm}^{a;b} &= \partial_b r_{knm}^a - i(A_{knn}^b - A_{kmm}^b)r_{knm}^a, \end{aligned} \quad (4)$$

where $|\partial_a u_{km}\rangle$ denotes the cell-periodic part of the Bloch eigenstate, and ∂_a is shorthand for $\partial/\partial k_a$.

It is crucial to note that, in calculations involving slab models, excluding the impact of the vacuum layer is essential for accurately determining the material’s effective shift current output, formalized as:

$$\sigma_{bb}^{a(\text{eff})} = \frac{l}{l_a} \sigma_{bb}^a, \quad (5)$$

where l represents the total thickness of the slab along the out-of-plane direction, and l_a denotes the material’s effective layer thickness. Specifically, for our slab under discussion, $l = 38.59\text{\AA}$ and $l_a = 13.26\text{\AA}$. For simplicity, the superscript ‘eff’ is omitted in subsequent diagrams and descriptions.

We implemented interpolation in k-space using the method of maximally localized Wannier functions (MLWFs), facilitated by the WANNIER90 code package^{23,24}. This technique was specifically employed to improve the convergence of SC conductivity calculations in k-space. The interpolated k-point grid was expanded to $500 \times 500 \times 1$. To specifically address the electronic properties near the Fermi level, trial orbitals for the Wannier projection were chosen as Ag-s and Te-p. The calculations of the aforementioned SC conductivity were conducted using the postw90-berry-sc module in the wannier90 post-processing program (postw90.x)²⁵.

The photon energy-dependent $\sigma_{bb}^a(\omega)$ in the slab across different in-plane components are shown in Fig. R9a (with Cartesian directions x and y labeled in the figure). The non-zero terms are σ_{xx}^y and σ_{yy}^y , both maintaining consistent signs and trends within the energy windows. The tiny value in σ_{xx}^x is attributed to algorithmic errors. For comparison, similar calculations were performed for bulk Ag_2Te (Fig. R9b), where all components consistently remain zero.

The calculation shows identical results with the qualitative symmetry analysis that a second-order nonlinear response along the b -axis exists on the surface of Ag_2Te , while prohibited in the bulk. The resembled σ_{xx}^y and σ_{yy}^y also predict a unipolar and relatively weak polarization dependence of the measured photocurrent. Phenomenologically, the calculation captures the symmetry and polarization characteristics in the observed SPGE.

On the other hand, we didn't observe the calculated peak near 1.28eV and 1.89eV. There is probably no such practical laser wavelength (Fig. 4b of the main text) that completely coincides with the predicted energy. Moreover, the prominent contribution from specific transition is likely covered in the more complex actual band structure, especially within the large energy range.

Figure R9 | Calculated shift current in the slab (a) and bulk (b) of Ag_2Te .

Comment 3:

How do temperature and laser modulation frequency affect the observed photocurrent? Exploring these properties may help reveal the origin of the photocurrent in this study.

Response:

We appreciate the suggestions about the two parameters. We tried our best to carry out the measurements and obtained some results as follows.

1) **As shown in Fig. R10, we measured the photocurrent within the liquid nitrogen temperature range in an optical dewar as well as the resistance.**

The photocurrent slightly fluctuates with the temperature decreasing from 300K to 120K, whereas a sharp drop emerges from 120K to 78K (Fig. R10c). Nonlinear optical response (especially shift current, which is independent of the carrier lifetime^{26,27}) is generally considered insensitive to temperature^{17,28} in case there is no significant change in absorption. In Ag₂Te, the excitation under visible illumination is unlikely to be critically affected by tuned bandgap at different temperatures. Besides, the elimination of the probable effect of Fermi level can be seen in response to the Comment 4 of Reviewer #2.

On the other hand, the two-terminal resistance of the device was also measured, which demonstrates a typical semiconductor behavior (Fig. R10d). The main transition temperature (~120K) is consistent in the photocurrent and resistance. **We attribute the drop in measured photocurrent to the effect of increased resistance, which induces more loss during the diffusion process of the photocurrent.** Similar temperature dependence is common in previous BPVE works^{16,17}.

We should emphasize that the photocurrent is generated at the surface of Ag₂Te. However, the contribution of resistance is global (from surface, bulk, and contacts) because the whole device participates in propagating the photocurrent.

Figure R10 | Temperature dependence of the photocurrent (low temperature). (a-b) Optical image (a), photocurrent mapping at room temperature (b), and corresponding reflection image (inset of a). (c) Temperature dependence of the photocurrent (absolute value). The data is obtained with illumination at the center of the device marked by the red spot in (a). (d) Temperature dependence of the two-terminal resistance. The laser wavelength is 690nm and the power is $\sim 14 \mu\text{W}$.

2) There is a structure phase transition in Ag_2Te from the monoclinic β -phase to the cubic α -phase at high temperature (Fig. R11b). The transition temperature is about 417K according to theoretical calculation²⁹. However, the high temperature exceeds the operation range of our optical dewar and we didn't observe signs of phase transition in the temperature-dependent measurement shown in the last part.

We tried to explore the phase transition employing a pair of micro heaters, as shown in Fig. R11a. The golden heaters were fabricated together with the sample electrodes. Such a single heater structure is common in thermoelectric transport measurements which can provide a thermal gradient. In our experiment, we applied a large voltage to the two heaters simultaneously to form a near-uniform temperature distribution.

As shown in Fig. R11d-f, the photocurrent slightly decreases with the heater voltage increasing from 0 to 10V and completely vanishes with the heater voltage of 18.5V. The photocurrent can re-emerge while the device cools down. **We attribute it to the structure phase transition. The α -phase in high temperature, belonging to**

space group Fm3m, is highly symmetric. The symmetry reduction at the surface related to the confined glide in β -phase is not allowed in α -phase.

We should state that this experiment is not quite mature. It's hard to obtain a definite temperature and check the uniformity of its distribution. The temperature of the substrate is roughly estimated as more than 400K, acquired by a mini thermometer in the atmospheric environment. Instead, the photocurrent measurement was carried out in a vacuum as an adiabatic environment. Moreover, the temperature of the sample close to the heaters should be higher than the whole substrate. On the other hand, our research about the α -phase is relatively inadequate. The phase transition lacks more electric and structural information. **However, as preliminary evidence of the phase transition, the experiment shows the significance of crystal structure in the generation of the photocurrent.**

Figure R11 | Temperature dependence of the photocurrent (high temperature). (a) Optical image of the device with a pair of heaters. (b) Reflection image of the sample. (c) The phase transition and corresponding crystal structure. (d-f) Photocurrent mapping with different heater voltage and corresponding heater power. The data in (f) is magnified as eight times. The laser wavelength is 690nm and the power is $\sim 70\mu\text{W}$.

3) We measured the mechanical chopper frequency dependence of the photocurrent (Fig.R12). **The response shows no decay in the range of 1~10kHz under both 690nm and 10.6μm illumination.** We are so sorry that our experiment condition is incapable of higher frequency measurements.

Based on the current results, we could evaluate an upper limit of the response time. According to the function of photocurrent I , frequency f and response time τ as $I = I_0 / \sqrt{1 + (2\pi f\tau)^2}$, **the response time in Ag₂Te should be no more than 3μs**, which would clearly decay in this frequency range. A short response time is expected due to the high mobility in Ag₂Te.

Mechanisms involving long response time could be excluded, such as the photogating effect. As for thermo-related processes, such as the bolometric (BOL) effect and photo-thermoelectric (PTE) effect, they were considered to have long response times in the past. Nevertheless, ultrafast thermal relaxation dynamics are

found to highly accelerate the thermal process in graphene and Dirac semimetals, leading to a short response time as several picoseconds^{30,31}. As a topological insulator with a narrow bandgap in the bulk, Ag₂Te is convinced to possess a relatively short transient lifetime, which would also benefit its response time in PTE. On the other hand, BPVE (or SPGE in our work) has an intrinsic ultrafast feature, due to the shift current generated during optical excitation^{30,32,33}. Consequently, we think it's challenging to distinguish SPGE from PTE by analyzing the time scale, even though we obtain a more accurate response time. However, PTE has been clearly excluded from our claim owing to the mirror-symmetric device structure. Additionally, PTE would hold its sign in the turn-over experiment, similar to BPVE, which differs from the measured SPGE.

Anyway, bandwidth is a crucial parameter for high-speed photodetector and optical communication. We look forward to further research about the bandwidth of Ag₂Te in the future.

Figure R12 | Frequency dependence of the photocurrent under 690nm (a) and 10.6μm (b) illumination.

Based on the reviewers' comments, we have performed the following corrections:

Revised manuscript:

1. Line 62, we have rewritten the BPVE expression to avoid confusion between β (in the experiment) and σ (in the calculation);
2. Lines 190-193, we have added a description of the theoretical calculation;
3. Line 214, we have added the calculation into consideration;
4. Lines 245-250, we have added a discussion about the strain effect.

Revised supplementary materials:

1. Figure S6c, we have modified the definition of polarization ratio from $(I_{\max}-I_{\min}) / (I_{\max}+I_{\min})$ to I_{\max}/I_{\min} ;
2. Text 7 and corresponding Figure S7, we have added the part 'Elimination of strain effect';
3. Text 8 and corresponding Figure S8, we have added the part 'Temperature dependence of the photocurrent';
4. Text 9 and corresponding Figure S9-10, we have added the part 'theoretical calculation of shift current'.

We would like to thank the reviewer for the critical comments and helpful suggestions. Indeed these comments inspire us to thoroughly analyze our data and come up with proper explanations and modeling. With the above response, we have addressed all the reviewer's questions. We hope that these analyses will address the reviewer's concerns and help to re-evaluate our manuscript. Thank you very much.

Response to Reviewer #2

General comment:

Photovoltaic effect reflecting the symmetry breaking of crystals is now one of the central topics in condensed matter physics due to its potential of overcoming the Shockley-Queisser limit in the conventional solar cells and its intrinsic mechanisms related with the carrier dynamics or band geometry/topology. Generally, non-centrosymmetric crystal structures is required to observe such intrinsic photovoltaic effect.

In this paper, authors reported the surface photogalvanic effect in centrosymmetric Ag_2Te . Although bulk crystal has a centrosymmetric crystal structure, inversion symmetry is broken at the surface, causing large photovoltaic response. They demonstrated that top and bottom surfaces generate the photocurrent of opposite sign and also clarified that photovoltaic response is activated across an ultrabroad spectral range from visible to mid-infrared light.

I think these findings provide a new direction of studying the photovoltaic effect and useful for applications in solar cells.

Response:

We thank the reviewer for the careful reading of our manuscript and we appreciate the positive and encouraging comments on our work. In the following, we would like to provide detailed answers to all the comments. The corresponding revisions have been implemented in the revised manuscript.

Comment 1:

It is difficult to intuitively understand how symmetry is reduced at the surface of Ag_2Te . Is there in-plane polarization on the surface? I think it may be good to draw polyhedra and bonds made up of re-adjacent atoms in the schematics (Fig. 1a, for example).

Response:

Thank you very much for your attention to the symmetry reduction. **We carefully checked the symmetry in Ag_2Te and found that we made a mistake in the original manuscript. The screw rotation symmetry C_{2b} is reserved and the glide mirror symmetry \widetilde{M}_b is broken at the surface.** We confused the two in the original manuscript. Our cognition about the crystallographic orientation of the nanoplate was limited in the past, which misled us to the wrong symmetry analysis. We have corrected this issue in the revised manuscript and we will explain it in detail in the following.

1) **We redraw the schematic with polyhedral and bonds for a more intuitive demonstration of the symmetry operations, as shown in Fig. R13.** There are three symmetry operations in the bulk of Ag_2Te , belonging to the space group $P2_1/c$ (No.14): twofold screw rotation symmetry with the axis C_{2b} along the crystallographic b -axis, glide mirror symmetry with the plane \widetilde{M}_b perpendicular to the crystallographic b -axis, and inversion symmetry, which is the product of the previous two operations. To exhibit

the microscopic operations, we focus on the (100) plane, where the b and c axes are completely in-plane (Fig. R13b). The two series of polyhedra, colored blue and red, show the necessary glides of $1/2b$ along the b -axis in the screw rotation symmetry and $1/2c$ along the c -axis in the glide mirror symmetry, respectively.

The glide operation is found significant in the symmetry reduction at the surface. The surface of the nanoplate is $(\bar{1}01)$ plane, where the c -axis is mostly out of the plane and the b -axis is completely in plane. **As a result, the translational symmetry along the c -axis is broken at the surface, and the glide mirror symmetry \widetilde{M}_b is broken at the surface as well. In contrast, the glide along the b -axis is insusceptible so the screw rotation symmetry C_{2b} is preserved at the surface.** The space group and point group of the surface would be $P2_1$ and C_2 , respectively. Correspondingly, the non-vanishing elements of β are β_{yxx} , β_{yyg} and β_{xxy} .

Figure R13 | Crystal structure of Ag_2Te . (a) Unit cell of Ag_2Te . The inky blue axis, the orange-pink plane, and the black dot indicate the screw rotation axis C_{2b} , the glide mirror plane \widetilde{M}_b and the center of inversion symmetry, respectively. The colored axes show the unit cell vectors. The dotted black axes show the laboratory coordinates, where the nanoplate surface $(\bar{1}01)$ is parallel to the x - y plane. (b) The (100) plane of the unit cell. The blue-outlined polyhedron transforms to the dotted blue one after the operation of a pure rotation and then coincides with the yellow one after a glide of $1/2b$ along the b -axis. The red-outlined polyhedron transforms to the dotted red one after the operation of a pure mirror and then coincides with the yellow one after a glide of $1/2c$ along the c -axis.

2) Previously the b -axis was recognized as aligned with one of the edges of the nanoplate according to scanning transmission electron microscopy (STEM)⁸. However, the corrected symmetry analysis leads to a new case that the b -axis should be perpendicular to the edge2 of the sample shown in Fig. 2 of the main text. **We performed many electric and optical anisotropy measurements and finally verified that there are two possible crystallographic orientations of the nanoplates. The b -axis is parallel or perpendicular to one of the edges of the nanoplates.** The chemical

vapor deposition (CVD) environment is complicated and it's conceivable that samples of different batches sometimes grow referring to different orientations: $\langle 101 \rangle$ or $\langle 010 \rangle$, marked as x or y in Fig. R13a, respectively. In fact, both of the two are the high symmetric orientations due to the C_{2b} and \bar{M}_b .

Next, we will show the evidence of the two crystallographic orientations that are developed spontaneously during CVD growth.

2.1) The vanishing of the surface photogalvanic effect (SPGE) induced photocurrent in this work is a clear probe for the crystallographic orientation.

According to the corrected symmetry analysis, the photocurrent perpendicular to the b -axis would vanish under unpolarized light. Therefore, the b -axis in the nanoplate shown in Fig. 2 of the main text, or Fig. R14a-b here, should be perpendicular to one of the edges. We performed the same experiments on many nanoplates and found there is another case, as shown in Fig. R14c-d. The photocurrent vanishes in s3, which indicates the b -axis is parallel to one of the edges in this nanoplate.

Figure R14 | Photocurrent in Ag₂Te nanoplates with different b -axis orientations.

(a,b) Photocurrent in a nanoplate where the b -axis is perpendicular to the edge, copied from Fig.2 of the main text. The photocurrent in s3 is vanished, which is parallel to the edge. (c,d) Photocurrent in a nanoplate where the b -axis is parallel to the edge. The photocurrent in s3 is vanished, which is perpendicular to the edge. The red dotted arrows in (a,c) mark the directions of the measurements in (b,d). The original shapes

of the nanoplates are outlined by white dotted lines in (a,c). The scale bars are 10 μm . The laser wavelength is 690nm (c,d) and the power is $\sim 100 \mu\text{W}$ (c) and $\sim 70 \mu\text{W}$ (d).

2.2) We measured the in-plane anisotropy of the conductivity in Ag_2Te nanoplates (Fig. R15). The four-terminal transport measurement in a disc-shape sample is highly reliable and it can exactly reflect the intrinsic property of the crystal. There are two cases that the maximum conductivity can orientate perpendicular or parallel to one of the edges of the nanoplates, which also implies two orientations of b -axis.

Figure R15 | Anisotropy of the conductivity (four-terminal) in Ag_2Te nanoplates with different b -axis orientations. The nanoplates are etched into disc shape for accurate measurement. (a) The maximum conductivity orientates along y axis, which is nearly perpendicular to the short edge at the bottom right of the image. (b) The maximum conductivity orientates along the edge e1. (b) is copied from reference⁸.

2.3) We measured the polarization-dependent absorption spectrum in different nanoplates (Fig.R16) using a Fourier transform infrared spectrometer (FTIR). Detailed information about the absorption measurement can be seen in the response to Comment 2 of Reviewer #1. The results also show two cases in which the orientation of maximum absorption can be perpendicular or parallel to one of the edges of the nanoplates.

Note that there is no actual edge along the 90° direction in the nanoplate shown in Fig.R16a. However, in the parallelogram shape nanoplates, the 90° direction is also a potential choice for the formation of edge. A new edge along this orientation is possible to appear after mechanical crushing, just like Fig. 3e-g of the main text. We cognize that this orientation as well as the other two edges are equivalent high-symmetric orientations in the morphology. Here we follow the expression of edge for convenience and call it ‘the third edge’.

Figure R16 | Polarization-dependent absorption spectrum of Ag_2Te nanoplates with different b -axis orientations. The maximum absorption is at 0° in (a) and at 90° in (b). The direction of 90° is along ‘the third edge’ of the nanoplates.

In summary, we claim that there are two possible orientations of b -axis through photocurrent, conductivity, and absorption measurements. Generally speaking, the b -axis is parallel or perpendicular to one of the edges of the nanoplates. We should explain that it’s challenging to employ the most common method, Raman spectrum, to distinguish the crystallographic orientation because Ag_2Te is susceptible to large-intensity illumination. However, as an optical process, the vanishing of SPGE is also a strong identification of the crystallographic orientation.

3) **There exists potential in-plane polarization on the surface of Ag_2Te according to the symmetry constraint. We performed a calculation about the polarization.**

In the revised manuscript, we constructed a slab model to calculate the nonlinear optical conductivity of the surface of Ag_2Te nanoplate. **Here, we would use the slab model to estimate the polarization of the surface as well.** Details about the slab (the setup of the structure and the DFT calculation process, along with the selection of general parameters) are shown in the revised Supplementary Information or the response to Comment 2 of Reviewer #1.

We employed the Wannier Center (WC) method in the polarization calculation, noted for its clear physical depiction²³. This method decomposes the total polarization vector \mathbf{P} into ionic and electronic contributions, formulated as:

$$\mathbf{P} = \mathbf{P}_{ion} + \mathbf{P}_{el} = \frac{e}{\Omega} \left[\sum_{n=1}^{N_{ion}} Z_n \mathbf{r}_n - \sum_{m=1}^{N_w} g_m^{(s)} \langle \mathbf{r} \rangle_{W_m} \right], \quad (6)$$

where N_{ion} (N_w) denotes the number of positive (negative) charge centers, each ion (WC) at position \mathbf{r}_n ($\langle \mathbf{r} \rangle_{W_m}$) with positive (negative) charge $+eZ_n$ ($-eg_m^{(s)}$). In this equation, Ω represents the volume of the cell, e is the elementary charge, Z_n is the ion charge number, and $g_m^{(s)}$ represents the degeneracy of electrons associated

with the WC.

It is important to recognize that the Z_n corresponds to the number of valence electrons considered for each individual atom in Density Functional Theory (DFT) calculations, and N_w represents the total number of WCs. The balance between these quantities is expressed as:

$$\sum_{n=1}^{N_{\text{ion}}} Z_n = \sum_{m=1}^{N_w} g_m^{(s)}. \quad (7)$$

For atom Ag, the valence electrons considered total 11 ($4d^{10}5s^1$) and for atom Te, they total 6 ($5s^25p^4$). From a chemical bonding perspective, the 5s electrons of Ag are used to fill the vacant 5p orbitals of Te, resulting in fully occupied orbitals of Ag (4d) and Te (5s5p). Accordingly, Wannier projectors were set to the Ag-d and Te-sp orbitals. The Wannier projection process and the calculation of the Wannier centers were conducted using the Wannier90 code package²⁴. The calculated results of the polarization vector \mathbf{P} are documented in Table II.

Table II| Calculated polarization for slab Ag₂Te.

Polarization (C/m^2)	x -direction	y -direction	z -direction
\mathbf{P}_{ion}	6.9492012	12.0920216	44.9324633
\mathbf{P}_{el}	-6.9492290	-12.0944405	-44.9324757
$\mathbf{P} = \mathbf{P}_{\text{ion}} + \mathbf{P}_{\text{el}}$	-0.0000279	-0.0024189	-0.0000124

We emphasize that the treatment of polarization in periodic systems typically requires identifying a reference structure (commonly one with an inversion center or an opposite polarization). It is crucial to track changes in polarization during structural transformations and subtract any discontinuities associated with these changes. In the context of WCs, such discontinuities arise when atoms cross the cell boundaries. We will next demonstrate that for the selected slab structure, the calculated polarization vector satisfies $\mathbf{P} = \frac{1}{2}\Delta\mathbf{P} = \frac{1}{2}(\mathbf{P} - \mathbf{P}^{\text{inverse}} - \mathbf{P}^{\text{mutation}})$, where $\Delta\mathbf{P}$ is the difference in polarization vectors between the slab itself and its inverted counterpart, excluding the polarization discontinuities. The inversion operation to construct the counterpart structure was performed using the slab cell center as the inversion center, ensuring that no atoms are displaced outside the cell. Consequently, it is unsurprising that the polarization vector of the inverse counterpart structure $\mathbf{P}^{\text{inverse}} = -\mathbf{P}$. Subsequently, we indexed all atoms of both structures according to their positions along the c -axis and tracked their movements during the linear transition from the slab to its inverse counterpart. We have monitored each marked atom and confirmed that no atoms crossed the cell boundaries during this transition. Thereby the polarization mutation, $\mathbf{P}^{\text{mutation}}$, is verified to be zero.

Now we could focus on the results in Table II, which shows that **the polarization along y -direction has a distinct value rather than x and z components. It is expected, because y (b) is the screw rotation axis preserved at the surface, or**

rather, polar axis.

We should emphasize that the symmetry constraints for electric polarization and bulk photovoltaic effect (BPVE) are not strictly consistent. However, BPVE has been firstly and commonly observed in ferroelectric materials. Recent work¹ shows polarization can significantly enhance the same-directional BPVE. **The in-plane polarization at the surface of Ag₂Te is likely to play a similar role in enhancing the observed SPGE.** Moreover, we look forward to further research on the surface polarization in Ag₂Te.

Comment 2:

I think surface photogalvanic effect can in principle occur in many centrosymmetric materials but there have been a few reports so far. What is special for Ag₂Te? Does the topological surface state (high mobility carriers only near the surface or quantum geometric nature of the surface carriers) play the important role?

Response:

This question is quite worth discussing, which involves the universality of SPGE. **First, we should declare that the SPGE observed in this work doesn't rely on the topological properties.** Our major experiments were performed under visible and near-infrared light, where the excitation energy is rather large. The transitions would be far from the Dirac cone. Therefore, the contribution to SPGE generation of Dirac fermion should be negligible. The theory of SPGE is mainly based on symmetry reduction and doesn't depend on topological properties, either.

1) A key point for the observed SPGE in Ag₂Te is the symmetry. Among centrosymmetric materials, Ag₂Te has a quite low symmetric monoclinic structure (P2₁/c space group). The symmetry operations are brief, just consisting of a screw rotation symmetry, a glide mirror symmetry, and their product, inversion symmetry. The breaking of anyone here would lead to the non-vanishing coefficient β_{qrs} . On the other hand, the surface of Ag₂Te nanoplate is not the highest symmetric (100) plane (shown in Fig. R13b). Therefore, the glide mirror symmetry is broken due to the out-of-plane glide along the *c*-axis.

Instead, a great number of well-studied centrosymmetric materials refer to high symmetry:

Some of them hold in-plane two-fold rotation symmetry C_{2z} (for example, graphene), which directly leads to the theoretical vanishing of β even though at the surface.

Some others involve in-plane three-fold rotation symmetry C_{3z} . Though C_{3z} doesn't prohibit the existence of β in principle, the experimental results show the response is usually weak. For example, the monolayer of 2H-WS₂, which is non-centrosymmetric, could be considered a low-symmetric surface of bulk 2H-WS₂ according to our theory of SPGE. However, recent work¹⁷ shows the bulk photovoltaic effect (BPVE) in monolayer WS₂ is not observable. So, the potential SPGE in bulk 2H-WS₂ is also expected too weak to be observed. The weakness is not confined by its trivial topologic property but an issue of symmetry, which could be largely enhanced by means of a lower-symmetric nanotube structure¹⁷. In fact, the weakness in materials with C_{3z} is also observed even in non-centrosymmetric structures such as 3R-MoS₂, which could be enhanced via breaking the C_{3z} by an applied strain¹. **We suggest any in-plane rotation symmetry would clearly suppress BPVE or SPGE.** As we discussed in response to Comment 1, in-plane rotation symmetry prohibits in-plane electric polarization, which can benefit BPVE or SPGE.

The glide is also significant in the symmetry reduction in Ag_2Te . As a comparison, many of the symmetry operations of the well-known anisotropic optical material black phosphorus (BP) would be broken on the surface. However, a two-fold rotation symmetry C_{2x} , a mirror symmetry M_x , and their production, inversion symmetry, would be preserved. **Though it seems like the macroscopic symmetry of Ag_2Te , the absence of the glide in BP makes it stable under the surface boundary confinement.**

Another similar case is $1T'$ - MoTe_2 , which belongs to $P2_1/m$ space group. A glide operation exactly exists in this structure. However, the glide direction is completely in-plane, making it stable under the surface boundary confinement.

In conclusion, suitable symmetry for SPGE is not as common as our instinct. A quite low symmetry among non-centrosymmetric structures is necessary. Or rather, research on the surface of a high Miller index is another approach to reduce the symmetry. This approach is feasible in three-dimensional materials and demands a particular growth environment or cut-plane technique. Instead, the naturally formed surface in low-dimensional materials would be relatively high-symmetric.

2) A sufficient thickness is also necessary to observe the intrinsic SPGE if there is no extra effect to break the symmetry of the top and bottom surfaces. Sometimes potential SPGE could be missed in research of two-dimensional materials for people often prefer to use thinner samples through mechanical exfoliation.

3) **The excellent transport property⁷ benefits the measured SPGE in Ag_2Te .** Under visible illumination, the SPGE is generated at the surface with excitation far from the Dirac cone. However, as a ‘non-local’ photocurrent (the illuminated region is far from contacts), its diffusion and collection also affect the measured result, which is participated by both surface and bulk states. The high mobility guarantees a relatively low loss in diffusion.

On the contrary, the photocurrent measurement would be more difficult in semiconductors with a large band gap. Nevertheless, their merit is the condition with a bias voltage, whereas the optical response would be covered by a large dark current in Ag_2Te .

4) **According to the non-linear optical theory, the topological properties can provide potential enhancement to SPGE.** The nontrivial Berry phase near the Dirac cone in Ag_2Te ³⁴ would show an outstanding contribution to the shift current within a low excitation energy range.

We performed SPGE measurement with laser wavelength up to $10.6\mu\text{m}$ as shown in Fig. 4b of the main text. The Dirac cone should provide a relatively dominant contribution to the results in the mid-infrared range. However, as we explained in the main text, it’s not rigorous to treat the wavelength-dependent results as spectrum analysis due to the difference in laser spots and optical systems. Instead, Fig.4b is more like a demonstration of the performance of the SPGE device. A defined study on the enhancement of topological property to SPGE is opening.

On the other hand, circular polarization-related measurement within mid/far-infrared range, or even the terahertz range, is rather helpful to distinguish the topological properties. We look forward to further investigations in Ag₂Te with circular polarization and low energy.

Comment 3:

Related with the above question, I would like to know the authors' opinion on the possible origin of the observed surface photogalvanic effect. Do authors think that something similar to the bulk photovoltaic effect (shift current, ballistic current, injection current etc.) are happening on surfaces?

Response:

We completely understand the concern about the microscopic mechanism of the observed SPGE. As a lowered-dimensional phenomenon of BPVE, the SPGE should have the same framework as BPVE. In the following, we will adopt the theory of BPVE.

In the original manuscript, we couldn't distinguish the mechanism between shift current and ballistic current based on the experimental results. In the revised manuscript, we add a numerical calculation about shift current and find it consistent with the experiment.

1) At first, we define the different kinds of mechanisms here²⁷.

Considering second-order perturbation under linear polarized illumination, the steady-state photocurrent can be derived as two parts: the off-diagonal part and the diagonal part of the density matrix. **The off-diagonal part is shift current (linear)**, which can be described as follows:

$$j_{shift} = E_r E_s e \sum_{n,l} \int d\mathbf{k} I_{nl} R_{nl}, \quad (8)$$

where E_r and E_s are the light electric fields, e is the elementary charge, I_{nl} is the transition rate from band n to l , and R_{nl} is the shift vector, indicating a unit shift in real space. R_{nl} is related to the Berry connection, making the shift current related to quantum geometry.

On the other hand, the diagonal part of the photocurrent is typically zero in nonmagnetic systems unless additional scattering processes are taken into account, such as electron-phonon interaction, electron-hole interaction, and scattering from defects. **This term induced by scattering is known as ballistic current (linear)** and can be expressed as follows:

$$j_{ballistic} = e\tau_0 \sum_{c,v,\mathbf{k}} \Gamma_{cv,\mathbf{k}} [v_{c,\mathbf{k}} - v_{v,\mathbf{k}}], \quad (9)$$

where τ_0 is the carrier lifetime, Γ_{cv} is the carrier generation rate corresponding to the transition from valence band v to conduction band c , and v_c as well as v_v are the electron velocities. Clearly, the ballistic current demonstrates a photocurrent induced by asymmetric carrier generation.

In magnetic systems (time reversal symmetry \mathcal{T} broken), the ‘intrinsic’ diagonal part is non-zero, referring to the injection current (linear). This case is irrelevant to our work in Ag_2Te .

The above definitions start from linear polarized illumination. In the case of circular polarized illumination, the photocurrent density matrix could be derived to the off-diagonal part and diagonal part as well.

The off-diagonal part also refers to shift current (circular), which usually vanishes unless in magnetic systems (\mathcal{T} broken).

The ‘intrinsic’ diagonal part is non-zero in the circular case, termed as injection current (circular). Here is the more well-known injection current, often dominant in circular photogalvanic effect (CPGE).

The scattering-induced ballistic current (circular) also exists in the circular case.

In conclusion, the presence or not of the mechanisms is related to the symmetry and polarization states. **The generally called shift current and ballistic current refer to the linear case, and the generally called injection current refers to the circular case.**

2) Our work doesn’t involve circular polarization and magnetism. Therefore, we just consider shift and ballistic current here.

It’s hard to distinguish the two experimentally. One probable approach is to apply an external magnetic field to induce Hall effect of the ballistic current³⁵. However, on the one hand, we don’t have the experiment condition. On the other hand, the reliability of the treatment is still doubtful²⁷. The applied magnetic field, which breaks the \mathcal{T} symmetry, would induce a new photocurrent. Besides, the magnetic dependence of shift current also requires further confirmation.

However, we performed a numerical calculation of the SPGE based on shift current framework. More details are shown in the revised Supplementary Information or the response to Comment 2 of Reviewer #1.

As shown in Fig.R16, the calculation clearly shows the absence and presence of β in the bulk and surface of Ag_2Te , respectively. In the bulk of Ag_2Te , all the components are zero due to the centrosymmetry. Instead, photocurrent along y (b) axis appears on the surface. The calculated result is mainly consistent with our experimental result and symmetry analysis. The tiny value in β_{xxx} is attributed to algorithmic errors.

Then we pay attention to the spectrum behavior. **The calculation shows an important feature that the tendencies of β_{yxx} and β_{yyy} are mostly identical.**

This result corresponds to the weak polarization dependence in experiments. It’s convincing that the coherence of β_{yxx} and β_{yyy} is contributed by the joint density of states (JDOS) and the in-plane optical anisotropy in Ag_2Te is relatively weak. More detailed discussion about the polarization feature could be referred to in the response to

Comment 2 of Reviewer #1.

In short, the calculation of shift current reproduces the symmetry and polarization features in the experiment. Therefore, we suggest that shift current has a dominant contribution to the observed SPGE in Ag₂Te.

Figure R16 | Calculated shift current in the surface (a) and bulk (b) of Ag₂Te. Here slab marked in (a) represents an isolated surface structure aligned as $(\bar{1}01)$ plane.

Comment 4:

Is the bulk part completely unrelated to this phenomenon? If the carrier number changes in the bulk part, will it affect the surface photogalvanic effect?

Response:

Transport carrier (near Fermi level) is almost unrelated to the generation procedure of BPVE (or SPGE). Instead, the photon-excited carrier determined by JDOS plays an important role. In the major experiments of our work under visible and infrared illumination, even the transport carrier of the surface state has negligible contribution to the generation procedure due to the large excitation energy.

In the practical measurements as a ‘non-local’ photocurrent (the illuminated region is far from contacts), both the surface and the bulk participate in the diffusion and collection of the SPGE response. The transport properties of the whole sample contribute to this process.

1) **We measured the temperature-dependent transport properties to explore their effect.**

All the Ag₂Te nanoplates used in our experiments are n-type at room temperature. Part of them maintain n-type and others approach charge neutral point according to different Fermi levels. Here we demonstrate the typical behaviors of the two types in Fig. R17. And the temperature dependence of photocurrent in Ag₂Te mainly shows a sharp drop from 120K to 78K accompanied by the changed resistance (as shown in Fig. R10 in the response to Comment 3 of Reviewer #1).

Regardless of the two types of samples, the carrier density clearly decreases while cooled down from 300K. On the other hand, mobility is promoted due to the suppression of phonon-scattering. **Neither the carrier density nor mobility is consistent with the photocurrent. Instead, the whole resistance (the combined action of them as well as the contact resistance), affects SPGE as shown in Fig. R10 due to the process of the photocurrent propagating.**

As we discussed with Fig. R10, nonlinear optical response (especially shift current, which is independent of the carrier lifetime^{26,27}) is generally considered insensitive to temperature in case there is no significant change in absorption^{17,28}. Change in bandgap at different temperatures is negligible in our experiment at the visible illumination.

Figure R17 | Typical temperature-dependent transport behaviors in Ag_2Te . (a-c) A representative sample that maintains n-type at low temperature. (d-f) Another representative sample close to the charge neutral point at low temperatures. (a-f) Magnetic field dependence of the longitudinal resistance (a,d) and Hall resistance (b,e) at different temperatures. Temperature dependence of calculated carrier density and mobility (c,f). Two-carrier model fitting is performed on (e) within the range of 60~140K to obtain the carrier density and mobility as shown in (f). The transition in Hall resistance (e) at a lower temperature <60K is attributed to the anomalous Hall effect and is not processed.

2) Another issue should be noted that the change in carrier density also induces the movement of the Fermi level. To confirm the effect of the Fermi level, we performed a calculation about JDOS as shown in Fig. R18.

The scope of $\pm 1.0\text{eV}$ is sufficiently large in practical experiments. The calculated result shows there is almost no difference in transitions of n-type samples with different Fermi levels, even in the most extreme case which approaches the charge

neutral point at low temperature. Consequently, we don't observe a relationship between the photocurrent and carrier density.

Figure R18 | Calculated JDOS with different Fermi levels of the Ag₂Te slab model.

Note that we don't discriminate the surface and bulk in the discussion about carrier density, mobility, and absorption. On the one hand, the distinction between the surface and bulk in the experiment is nearly impossible, except for extremely low temperatures (where the topological surface transport properties could be identified through quantum oscillations). On the other hand, these factors at surface and bulk would change together even with similar trends. (Our experiment doesn't involve low photon energy comparable to the bandgap.) **However, as we stated before, the effect of transport properties mainly works in the photocurrent propagating process, which includes the contribution from surface and bulk in the meantime. While limited in the photocurrent generating process, the main effect of carrier density would be the Fermi level-induced change in absorption, which is proved to be negligible in this Ag₂Te work.**

Based on the reviewers' comments, we have performed the following corrections:

Revised manuscript:

1. Line 113, Table I, Fig. 1a-b caption, we have unified the glide mirror symbol as \widetilde{M}_b instead of M_b ;
2. Line 115 and Fig. 1a-b caption, we have added a description of the glide operation;
3. Line 118, we have corrected the information about the in-plane orientation of the b -axis;
4. Lines 181-186, we have added an explanation about the symmetry reduction;
5. Lines 188-207, we have corrected the symmetry analysis and adjusted the sequence of discussion about the crystallographic orientation, polarization, and the added theoretical calculation;
6. Fig. 2d caption, we have corrected the β elements related to the symmetry analysis;
7. Line 227 and Fig. 2f caption, we have deleted the description about electric polarization in BPVE for more universal;
8. Line 266, we have corrected the indication of the b -axis;
9. Line 338, we have added a discussion about the effective approach for symmetry reduction.

Revised figure and table:

1. In Table I, we have corrected the symmetry analysis and replaced the point group with the space group;
2. Fig. 1a-b, we have renewed the schematic of the crystal structure as a unit cell with polyhedral and bonds for a more intuitive demonstration of the symmetry operations;
3. Fig. 2a,d,e, we have corrected the indication of the b -axis and the symmetry analysis;
4. Fig. 2f, we have deleted the arrow indicating the electric polarization;
5. Fig. 3e, we have deleted the wrong indication of the b -axis.

Revised supplementary materials:

1. Text 2 and corresponding Fig. S3, we have replaced the old device (as a repeatable result of the crystal orientation dependence) with the one shown in Fig. R14 where the b -axis is parallel to the edge, and added the illustration about the two possible orientations of b -axis;
2. Line 120, we have corrected the β elements related to the symmetry analysis and deleted the speculation about b -axis orientation;
3. Line 128 and the title of Text 6, we have corrected the β elements related to the symmetry analysis;
4. We have deleted the part about the discussion for probable microscopic mechanism,

which leaves an open question in the original manuscript; instead, we have added the new part as Text 7 about the shift current calculation.

Overall, we thank all the Reviewers for their detailed comments and valuable suggestions. We have made the necessary revisions based on these comments and hope that our response is satisfactory to the reviewers. We also appreciate the significant effort that the reviewers have dedicated to this manuscript and the dramatic help to further improve our manuscript. Thank you all.

References

1. Dong, Y. *et al.* Giant bulk piezophotovoltaic effect in 3R-MoS₂. *Nature Nanotechnology* **18**, 36–41 (2023).
2. Ma, C. *et al.* Intelligent infrared sensing enabled by tunable moiré quantum geometry. *Nature* **604**, 266–272 (2022).
3. Otteneder, M. *et al.* Terahertz Photogalvanics in Twisted Bilayer Graphene Close to the Second Magic Angle. *Nano Lett.* **20**, 7152–7158 (2020).
4. Arora, A. Strain-induced large injection current in twisted bilayer graphene. *PHYSICAL REVIEW B* (2021).
5. Jiang, J. *et al.* Flexo-photovoltaic effect in MoS₂. *Nat. Nanotechnol.* **16**, 894–901 (2021).
6. Yang, M.-M., Kim, D. J. & Alexe, M. Flexo-photovoltaic effect. *Science* **360**, 904–907 (2018).
7. Leng, P. *et al.* Gate-Tunable Surface States in Topological Insulator β -Ag₂Te with High Mobility. *Nano Lett.* **20**, 7004–7010 (2020).
8. Chen, S. *et al.* Real-space observation of ultraconfined in-plane anisotropic acoustic terahertz plasmon polaritons. *Nature Materials* **22**, 860–866 (2023).
9. Schnell, M., Carney, P. S. & Hillenbrand, R. Synthetic optical holography for rapid nanoimaging. *Nat Commun* **5**, 3499 (2014).
10. Zhou, Y. *et al.* Giant intrinsic photovoltaic effect in one-dimensional van der Waals grain boundaries. *Nat Commun* **15**, 501 (2024).
11. Matsuo, H. *et al.* Bulk and domain-wall effects in ferroelectric photovoltaics. *Phys. Rev. B* **94**, 214111 (2016).
12. Ji, W., Yao, K. & Liang, Y. C. Evidence of bulk photovoltaic effect and large tensor coefficient in ferroelectric BiFeO₃ thin films. *Phys. Rev. B* **84**, 094115 (2011).
13. Knoche, D. S., Steimecke, M., Yun, Y., Mühlenbein, L. & Bhatnagar, A. Anomalous circular bulk photovoltaic effect in BiFeO₃ thin films with stripe-domain pattern. *Nat Commun* **12**, 282 (2021).
14. Wei, J. *et al.* Zero-bias mid-infrared graphene photodetectors with bulk photoresponse and calibration-free polarization detection. *Nat Commun* **11**, 6404 (2020).
15. Wei, J., Xu, C., Dong, B., Qiu, C.-W. & Lee, C. Mid-infrared semimetal polarization detectors with configurable polarity transition. *Nat. Photon.* **15**, 614–621 (2021).
16. Akamatsu, T. *et al.* A van der Waals interface that creates in-plane polarization and a spontaneous photovoltaic effect. *Science* **372**, 68–72 (2021).
17. Zhang, Y. J. *et al.* Enhanced intrinsic photovoltaic effect in tungsten disulfide nanotubes. *Nature* **570**, 349–353 (2019).
18. Liang, Z. *et al.* Strong bulk photovoltaic effect in engineered edge-embedded van der Waals structures. *Nature Communications* **14**, 4230 (2023).
19. Kresse, G. & Furthmüller, J. Efficient iterative schemes for *ab initio* total-energy calculations using a plane-wave basis set. *Phys. Rev. B* **54**, 11169–11186 (1996).
20. Perdew, J. P., Burke, K. & Ernzerhof, M. Generalized Gradient Approximation

- Made Simple. *Phys. Rev. Lett.* **77**, 3865–3868 (1996).
21. Blöchl, P. E. Projector augmented-wave method. *Phys. Rev. B* **50**, 17953–17979 (1994).
 22. Sipe, J. E. & Shkrebtii, A. I. Second-order optical response in semiconductors. *Phys. Rev. B* **61**, 5337–5352 (2000).
 23. Marzari, N., Mostofi, A. A., Yates, J. R., Souza, I. & Vanderbilt, D. Maximally localized Wannier functions: Theory and applications. *Rev. Mod. Phys.* **84**, 1419–1475 (2012).
 24. Pizzi, G. *et al.* Wannier90 as a community code: new features and applications. *J. Phys.: Condens. Matter* **32**, 165902 (2020).
 25. Ibañez-Azpiroz, J., Tsirkin, S. S. & Souza, I. *Ab initio* calculation of the shift photocurrent by Wannier interpolation. *Phys. Rev. B* **97**, 245143 (2018).
 26. Xu, H., Wang, H., Zhou, J. & Li, J. Pure spin photocurrent in non-centrosymmetric crystals: bulk spin photovoltaic effect. *Nature Communications* **12**, 4330 (2021).
 27. Dai, Z. & Rappe, A. M. Recent progress in the theory of bulk photovoltaic effect. *Chemical Physics Reviews* **4**, 011303 (2023).
 28. Ma, J. *et al.* Unveiling Weyl-related optical responses in semiconducting tellurium by mid-infrared circular photogalvanic effect. *Nat Commun* **13**, 5425 (2022).
 29. Zhang, W. *et al.* Topological Aspect and Quantum Magnetoresistance of β -Ag₂Te. *Phys. Rev. Lett.* **106**, 156808 (2011).
 30. Liu, J., Xia, F., Xiao, D., García de Abajo, F. J. & Sun, D. Semimetals for high-performance photodetection. *Nat. Mater.* **19**, 830–837 (2020).
 31. Liu, C. *et al.* Silicon/2D-material photodetectors: from near-infrared to mid-infrared. *Light Sci Appl* **10**, 123 (2021).
 32. Tan, L. Z. *et al.* Shift current bulk photovoltaic effect in polar materials—hybrid and oxide perovskites and beyond. *npj Comput Mater* **2**, 1–12 (2016).
 33. Laman, N., Bieler, M. & van Driel, H. M. Ultrafast shift and injection currents observed in wurtzite semiconductors via emitted terahertz radiation. *Journal of Applied Physics* **98**, 103507 (2005).
 34. Leng, P. *et al.* Nondegenerate Integer Quantum Hall Effect from Topological Surface States in Ag₂Te Nanoplates. *Nano Lett.* **23**, 9026–9033 (2023).
 35. Burger, A. M. *et al.* Direct observation of shift and ballistic photovoltaic currents. *Sci. Adv.* **5**, eaau5588 (2019).

REVIEWERS' COMMENTS

Reviewer #1 (Remarks to the Author):

The authors have properly addressed the issues raised in the initial review process. The manuscript has been improved substantially in terms of clarity and quality. The manuscript is in good condition for publication.

Reviewer #2 (Remarks to the Author):

The authors performed additional measurements and further discussed the symmetry and observed photocurrents. The manuscript has been appropriately improved and I have no additional questions.